# Plume-ridge interactions: Ridge-ward versus plate-drag plume flow

**Fengping Pang[1], Jie Liao[1,2,3], Maxim D. Ballmer[4], Lun Li[1,2,3]**

[1]School of Earth Sciences and Engineering, Sun Yat-Sen University, Guangzhou 510275, China

[2]Guangdong Provincial Key Lab of Geodynamics and Geohazards, Guangzhou 510275, China

[3]Southern Marine Science and Engineering Guangdong Laboratory (Zhuhai), Zhuhai 519000, China

[4]Department of Earth Sciences, University College London, London, United Kingdom

Correspondence: Jie Liao (liaojie5@mail.sysu.edu.cn)

**Abstract**

The analysis of mid-ocean ridges and hotspots that are sourced by deep-rooted mantle plumes allows us to get a glimpse of mantle structure and dynamics. Dynamical interaction between ridge and plume processes have been widely proposed and studied, particularly in terms of ridge-ward plume flow. However, the effects of plate drag on plume-lithosphere and plume-ridge interaction remain poorly understood. In particular, the mechanisms that control plume flow towards vs. away from the ridge have not yet been systematically studied. Here, we use 2D thermomechanical numerical models of plume-ridge interaction to systematically explore the effects of (i) ridge spreading rate, (ii) initial plume head radius, and (iii) plume-ridge distance. Our numerical experiments suggest two different geodynamic regimes: (1) plume flow towards the ridge is favored by strong buoyant mantle plumes, slow spreading rates and small plume-ridge distances; (2) plume drag away from the ridge is in turn

promoted by fast ridge spreading, at least for small-to-intermediate plumes and large plume-ridge
distance. We find that the pressure gradient between the buoyant plume and spreading ridge at first
drives ridge-ward flow, but eventually the competition between plate drag and the gravitational force
of plume flow along the base of the sloping lithosphere controls the fate of plume (spreading towards
vs. away from the ridge). Our results highlight that fast-spreading ridges exert strong plate dragging
force, which sheds new light on natural observations of largely absent plume-lithosphere interaction
along fast-spreading ridges, such as the East Pacific Rise.

## 1   Introduction

Mid-ocean ridges (MORs) and hotspots are two main regions for deep material recycling to the surface of the Earth. However, these two units are not always isolated, but rather show strong interactions in some cases, termed as plume-ridge interaction (Morgan, 1978). Of up to 50 mantle plumes revealed by seismic tomography (French and Romanowicz, 2015; Montelli et al., 2004), more than 20 plumes are found to be associated with nearby ridges (Fig.1a; Ito et al., 2003). Plume-ridge interaction is manifested by geophysical and geochemical anomalies along the ridge axis, e.g., high mantle potential temperature (Dalton et al., 2014), enriched radiogenic isotope anomalies (Cushman et al., 2004; Douglass and Schilling, 1999; Yang et al. 2017), and adjacent lineations of seamounts (Ballmer et al., 2013b; Geissler et al., 2020; Lénat and Merle, 2009). Furthermore, plumes may promote migration of MOR spreading centers (Müller et al., 1998; Mittelstaedt et al., 2008, 2011; Whittaker et al., 2015), as evidenced by successive ridge jumps towards mantle plumes, e.g., at Iceland, Amsterdam-Saint Paul and Galapagos hotspots (Hardarson et al., 1997; Maia et al., 2011; Mittelstaedt et al., 2012). The interaction dynamics of a ridge with a ridge-centered and off-ridge plume has been widely studied and modeled in analogue and numerical experiments, revealing that the major controlling factors involve the ridge spreading rate, plume buoyancy flux and their spatial distance (François et al., 2018; Ito et al., 1997; Kincaid et al., 1996; Ribe et al., 1995; Ribe, 1996; Sleep, 1997). Indeed, most plume-ridge interaction systems are associated with slow-spreading ridges and small mantle plumes and short plume-ridge distances (Fig.1b). However, numerical studies systematically investigating the effects of these parameters on plume-ridge interaction and quantify the controlling forces remain scarce.

As has been noted previously, buoyant plumes tend to spread ridge-ward along the sloping base
of the lithosphere (Morgan, 1978; Schilling, 1991; Small, 1995). Regions of divergent mantle flow
beneath MORs represent the lowest dynamic-pressure regions in the oceanic asthenosphere, and thus
tend to suck ambient asthenospheric and plume materials towards the spreading center (Niu, 2014).
On the other hand, the viscous drag at the base of the plate tends to convey the spreading plume
material away from the MOR (Ribe and Christensen, 1994, 1999). Indeed, plume spreading at the
base of the lithosphere is governed by the competition of trench-ward viscous plate drag vs.
ridge-ward gravitational and pressure-driven forces (Kincaid et al.,1996). These gravitational and
tectonic forces compete with other to control the regime of plume-ridge interaction, but their balance
remains to be quantified.
The different distribution of hotspots with classified as plume-ridge interaction (ridge-ward
spreading) vs. no interaction (plate-drag spreading) also still remains enigmatic. Plume-ridge
interaction is much more common near the Mid-Atlantic ridge (MAR) than near the East Pacific Rise
(EPR) (Fig. 1a). Near the EPR, only the Pukapuka and Sojourn ridges display clear evidence of
ridge-ward flow of the magmatic source, but these volcanic ridges have been attributed to a
horizontally viscous differences or small-scale convection in uppermost mantle, and not a mantle
plume (Ballmer et al., 2013b; Clouard and Bonneville, 2005; Harmon et al., 2011). A previous study
(Jellinek et al., 2003) proposed that fast-spreading ridges guide upwelling mantle flow towards the
spreading center to convey the surrounding plumes from deep depth entirely into the MOR melting
zone (Fig. 1c), resulting in the absence of hotspots adjacent to the EPR (see also Rowley et al., 2016;
Rowley and Forte, 2022). However, fast plate spreading also tends to drag mantle plumes away from
the MOR (Kincaid et al., 1995, 1996), leading to the typically parabolic shapes of hotspot swells
such as near Hawaii (Ribe and Christensen, 1994). Whether the increased spreading rates in the
Pacific vs. Atlantic promote ridge-ward vs. plate-drag plume flow remains an intriguing question.
The principal goal of this study is to investigate the process of plume-ridge interaction, with an
emphasis on the effects of model parameters on the ridge-ward vs. plate-drag plume spreading. We
explore the effects of various model parameters, such as the size of the plume, ridge spreading rate,
and plume-ridge distance. Finally, we use our model results to interpret the difference of natural
plume-ridge interaction systems in different oceans, particularly the striking difference between the
East Pacific and Atlantic in this regard.

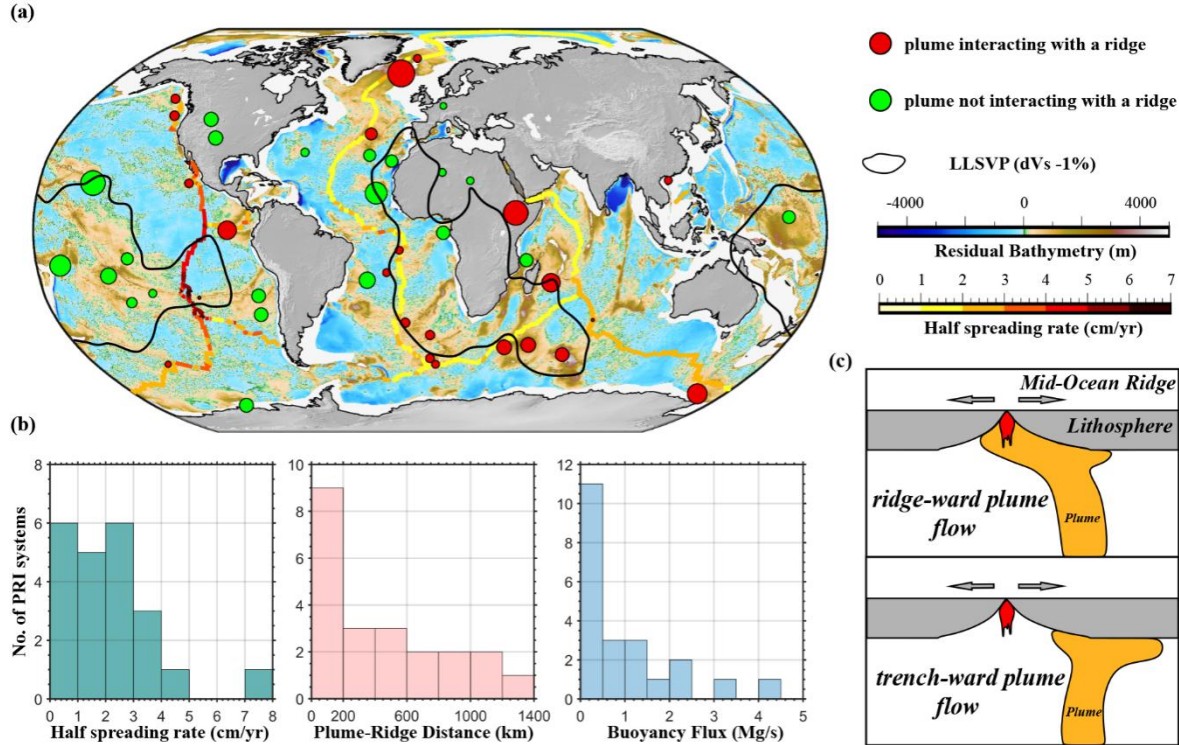

**Figure 1.** Global plume-ridge interaction systems. **(a)** Global distribution of mid-ocean ridges and mantle plumes. Residual bathymetry of the ocean basins come from Straume et al. (2019). Mid-ocean ridges are painted in color solid lines corresponding to half-spreading rate. Plumes not interacting with a ridge are shown by green circles, and hotspots linked to ridges are in red dots (Ito et al., 2003); size refers to the plume buoyancy flux from Hoggard (2020). Black lines denote the regions of two LLSVPs under the South Africa and Pacific Ocean (Torsvik et al., 2006). **(b)** Histograms of influential factors of plume-ridge interaction systems. Half spreading rate and plume-ridge distance is taken from GPlates (Müller et al., 2016; Whittaker et al., 2015). Plume-ridge interaction systems link to slow-spreading ridge and small mantle plumes and short plume-ridge distance. **(c)** Sketches of ridge-ward (top panel) and plate-drag plume flow (bottom panel) mode proposed, respectively.

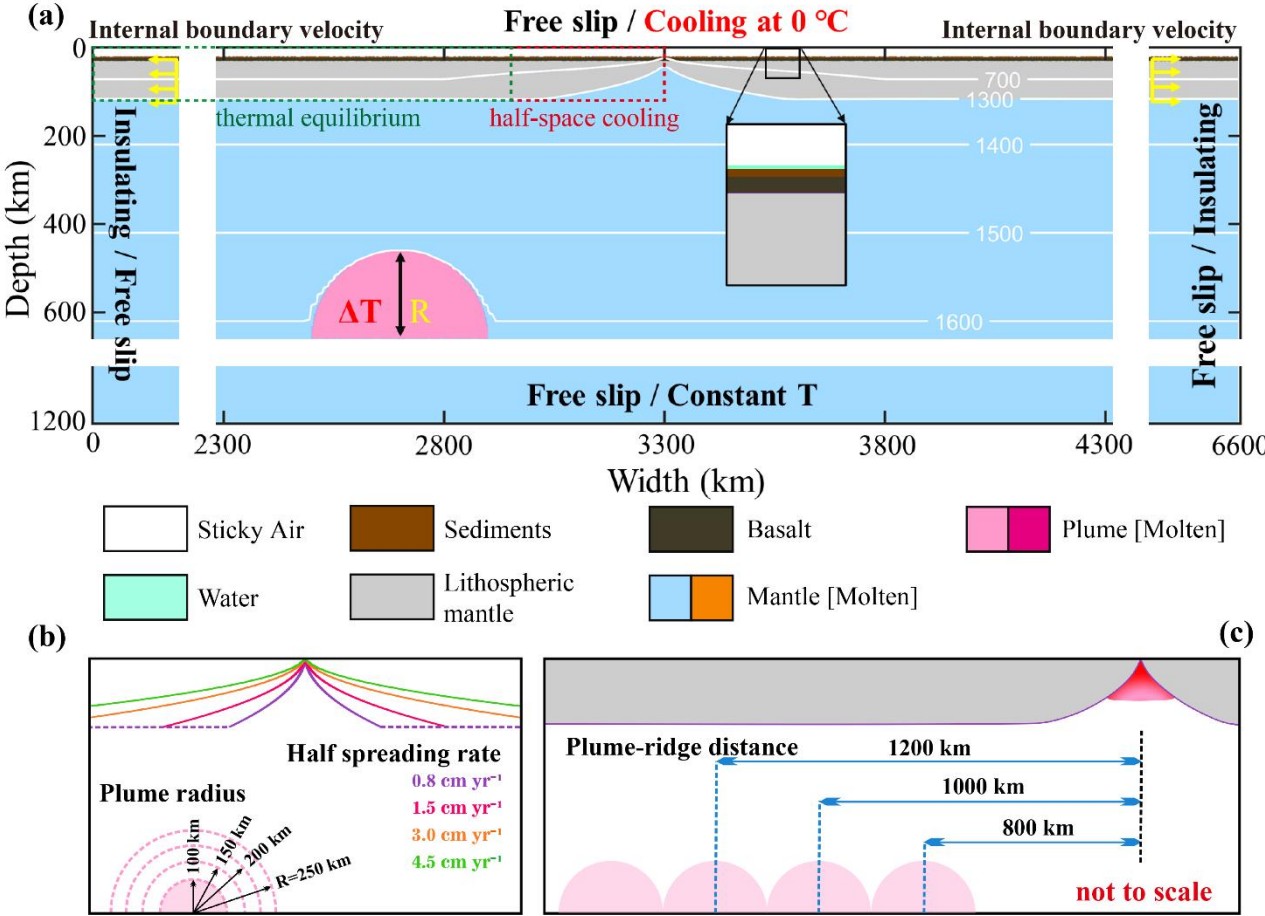

94

**Figure 2.** Model setup. **(a)** Initial composition and boundary conditions. The oceanic plate consists

of half-space cooling part and the thermal equilibrium part. A 50-Myrs-old mid-ocean ridge sets in

the middle of the model based on half-space cooling temperature structure. A thermal and chemical

anormal mantle plume locates at 660 km. Different colors indicate the initial rock types and

corresponding newly formed molten rock types. Yellow arrows are the half-spreading rates imposed

internal in the lithosphere (i.e., from 20 km to 120 km in depth) to simulate ridge spreading. **(b)**

Initial tested ridge and plume configurations. **(c)** Initial tested plume-ridge distances.

## 2 Numerical modelling

### 2.1 Modelling methods

To explore plume-lithosphere and plume-ridge interaction, we conduct numerical simulations

utilizing the 2D thermo-mechanical code I2VIS, which is based on staggered finite difference
method combined with marker-in-cell techniques (Gerya and Yuen, 2003, 2007). This modeling
framework uses both Eulerian grids and randomly-distributed Lagrangian markers to jointly solve
equations of conservation of mass, momentum and energy (Eq. (1)-(3), respectively):
$$\nabla \cdot \vec{v} = 0 \quad (1)$$

$$\frac{\partial \sigma'_{ij}}{\partial x_j} - \frac{\partial P}{\partial x_i} + \rho g_i = 0 \quad (2)$$

$$\rho C_p \left( \frac{DT}{Dt} \right) = - \nabla \cdot \vec{q} + H_r + H_a + H_s + H_l \quad (3)$$

where $v$ refers to the velocity, $\sigma'_{ij}$ the deviatoric stress tensor, $P$ the pressure, $\rho$ the density, $g$ the
gravity acceleration, $\frac{D}{Dt}$ the Lagrangian time derivative, $C_p$ the heat capacity, and $q$ the heat flux.
Additionally, $H_r$, $H_a$, $H_s$, and $H_l$ are the radioactive, adiabatic, shear, and latent heat productions,
respectively.
We employ a non-Newtonian visco-plastic rheology (Gerya and Yuen, 2007) in the models. The
viscous rheology depends on stress, temperature and pressure. The appropriate viscosity is expressed
as that of a composite diffusion and dislocation-creep material (Eq. (4)).
$$\frac{1}{\eta_{vis}} = \frac{1}{\eta_{diff}} + \frac{1}{\eta_{disl}} (4)$$

in which $\eta_{diff}$ and $\eta_{disl}$ are the diffusion and dislocation creep viscosity, respectively, and can be
further computed as Eq. (5) and Eq. (6):
$$\eta_{diff} = \frac{1}{2} A \sigma_{crit}^{1-n} \exp \left( \frac{PV_a + E_a}{RT} \right) \quad (5)$$

$$\eta_{disl} = \frac{1}{2} A^{\frac{1}{n}} \dot{\varepsilon}_{II}^{\frac{1-n}{n}} \exp \left( \frac{PV_a + E_a}{nRT} \right) \quad (6)$$

where $P$ is the pressure, $T$ is the temperature, $\dot{\varepsilon}_{II}$ is the second invariant of the strain rate tensor,
$\sigma_{crit}$ is the diffusion-dislocation creep transition stress, and $A$, $E_a$, $V_a$, and $n$ are the strain rate
pre-exponential factor, activation energy, activation volume, and stress exponent, respectively. The
plastic behavior $\eta_{pla}$ is described by the Drucker-Prager yield criterion (Byerlee, 1978; Ranalli,
1995) according to Eq. (7) and Eq. (8):

$$\sigma_y = C + P\varphi \quad (7)$$

$$\eta_{pla} = \frac{\sigma_y}{2\dot{\varepsilon}_{II}} \quad (8)$$

in which $\sigma_y$ is the yield stress, $C$ is the rock cohesion and $\varphi$ is the effective friction coefficient.
The effective viscosity $\eta_{eff}$ of rocks is thus constrained by both viscous and plastic deformation,
where the rheological behavior depends on the minimum viscosity attained between ductile and
brittle fields:

$$\eta_{eff} = \min(\eta_{vis}, \eta_{pla}) \quad (9)$$

Partial melting, melt extraction and percolation are also considered in the model in a simplified
way (Gerya, 2013). The melt fraction ($M_0$) of the crust are assumed to increase with temperature and
are calculated according to Eq. (10):

$$M_0 = 0 \text{ when } T \leq T_{solidus}$$

$$M_0 = \frac{(T - T_{solidus})}{(T_{liquidus} - T_{solidus})} \quad \text{when } T_{solidus} < T < T_{liquidus} \quad (10)$$

$$M_0 = 1 \text{ when } T \geq T_{liquidus}$$

where $T_{solidus}$ and $T_{liquidus}$ are the solidus and liquidus temperature of different rock types,
respectively, taken from Katz et al. (2003).
In our model, melt extraction is modeled indirectly and considered as an instantaneous process
(Gerya et al., 2015). The extracted melt is assumed to move vertically from the molten source and
then added to the bottom of the crust. Partial melt is extracted from the mantle and instantaneously
displaced to the bottom of the crust and converted into hot mafic magma, obeying the conservation
of material. The amount of extracted melt during the evolution of each experiment is traced by the
Lagrangian markers (Gerya, 2013). The total amount of melt, M, for every marker excludes the
amount of previously extracted melt according to Eq. (11):

$$M = M_0 - \Sigma_n M_{ext} \quad (11)$$

where $\Sigma_n M_{ext}$ refers to the total melt fraction extracted during the previous *n* melt extraction
timesteps.
The effective density of mafic magma and molten crust depends on its melt fraction and is
calculated as (Gerya et al., 2015; Gülcher et al., 2020):

$$\rho_{eff} = \rho_{solid}(1 - M + M\frac{\rho_{0,molten}}{\rho_{0,solid}}) \quad (12)$$

where $\rho_{0,molten}$ and $\rho_{0,solid}$ are the reference densities of the molten and solid crust. $\rho_{solid}$ is the
crust density at given pressure and temperature, which can be computed as:

$$\rho_{solid} = \rho_{0,solid}[1 - \alpha(T - 298)][1 + \beta(P - 0.1)] \quad (13)$$

with thermal expansion $\alpha = 3 \times 10^{-5} K^{-1}$ and compressibility $\beta = 10^{-11} Pa^{-1}$.
Surface processes, such as erosion and sedimentation, are considered by solving the transport
equation on the Eulerian nodes at each time step (Gerya and Yuen, 2003). Our erosion/sedimentation
model uses gross-scale erosion/sedimentation rates which are independent of local elevation and
topography (Burov and Cloetingh, 1997). We use constant and moderate rates of erosion (0.315
mm/yr) and sedimentation (0.0315 mm/yr), respectively, which falls within naturally observed
ranges.

**2.2 Model setup**
The size of the model box is 6600 × 1200 km, with a nonuniform grid of 501 × 301
computational nodes in length and depth, respectively (Fig. 2). The densest grid is located in the

center of the model domain (i.e., grid size decreases linearly from 20 km at the edges to 2 km at the

ridge axis), where plume-ridge interaction would happen. The model consists of a 20 km thick sticky

air layer to accommodate crustal surface deformation. To reproduce the oceanic lithosphere, we

choose a typical layered model, where the crust is composed of a water level (2 km), a sediment

layer (1.5 km), and a basalt layer (7.5 km). The oceanic lithosphere and asthenosphere in the model

are both modeled as dry olivine (the different colors for the mantle lithosphere and asthenosphere in

the figures of this paper are only for better visualization). Besides, a 50-Myrs-old mid-ocean ridge is

set on central part of the lithosphere, splitting the model domain into two parts. At the depth of 660

km, a 200-km-wide semicircular plume is located on the left of model domain, corresponding to the

onset of plume-ridge interaction from the mantle transition zone. Detailed rock parameters are listed

in Table 1.

The thermal conditions at the top and bottom boundaries are fixed at 273 and 2513 K,

respectively. The left and right boundaries are both insulating, with no external heat flow across them.

The initial temperature structure of the mantle is adiabatic (0.5 K km$^{-1}$), which results in a

temperature at 660 km depth of 1843 K. The initial temperature structure of the oceanic plate

consists of half-space cooling part and thermal equilibrium part (Fig. 2a). The half-space cooling

model is used to describe the oceanic plate younger than 50 Myr, and the thermal equilibrium

structure is used to describe older oceanic parts. In other words, the thermal age of the lithosphere far

away from the ridge is fixed at 50 Myr with a constant plate thickness (i.e., ~100 km). The hot plume

is set a circular thermal and compositional (see Table 1) anomaly with an excess temperature of 250

K to trigger a rising thermochemical plume. All the velocity boundaries are free slip boundaries.

Additional internal boundary velocities are imposed at 500 km from each side boundary in the

lithosphere to maintain the imposed half spreading rate (Fig. 2a).

**Table 1**. Rock physical properties used in the numerical models.

| Parameters | Sediments | Ocean Crust | Mantle | Plume | Reference [a] |
|---|---|---|---|---|---|
| Flow law | Wet quartz | Basalt | Dry olivine | Wet olivine | |
| Preexponential factor $A$(Pa$^n$s) | $1.97 \times 10^{17}$ | $4.80 \times 10^{22}$ | $3.98 \times 10^{16}$ | $5.01 \times 10^{20}$ | 1 |
| Activation energy $E_a$(KJ mol$^{-1}$) | 154 | 238 | 532 | 470 | 1 |
| Activation volume $V_a$(J bar$^{-1}$mol$^{-1}$) | 0 | 0 | 1 | 0.8 | 1 |
| Exponent $n$ | 2.3 | 3.2 | 3.5 | 4 | 1 |
| Cohesion $C$(Pa) | $2 \times 10^7$ | $2 \times 10^7$ | $2 \times 10^7$ | $2 \times 10^7$ | 1 |
| Effective friction coefficient $\varphi$ | 0.6/0.3 | 0.6/0.3 | 0.6/0.3 | 0.6/0.3 | 1 |
| Density $\rho$(Kg m$^{-3}$) | 2600 | 3000 | 3300 | 3270 | 2 |
| Radioactive heating $H_r$(W m$^{-3}$) | $2 \times 10^{-6}$ | $2.2 \times 10^{-7}$ | $2.2 \times 10^{-8}$ | $2.5 \times 10^{-8}$ | 2 |

a: 1-(Ranalli, 1995), 2-(Turcotte and Schubert, 2014)
Other physical parameters used for all rocks include: gas constant $R$=8.314 J K$^{-1}$mol$^{-1}$, thermal
expansion $\alpha$=$3 \times 10^{-5}$ K$^{-1}$, compressibility $\beta$=$1 \times 10^{-11}$ Pa$^{-1}$, heat capacity $Cp$=1000 J kg$^{-1}$K$^{-1}$.

**3    Model Results**

We conduct a series of numerical experiments to investigate ridge suction versus plate drag acts

on plumes. The effects of three major model parameters (i.e., the spreading rate of mid-ocean ridge,
the initial plume head radius, and the plume-ridge distance) are systematiclly studied. The typical
dynamic evolution of models with ridge-ward vs. plate-drag plume flow are demonstrated.
**3.1 Model evolution with ridge-ward plume flow**

For models with dominant ridge-ward flow, the typical model evolution is shown in Fig. 3 (the

major model parameters used in this case are: the half spreading rate of 8 mm yr$^{-1}$, the initial plume
head radius of 200 km, and the off-axis distance of 800 km). In the early plume head stage, the
buoyant mantle plume rises up rapidly in a mushroom-like shape (Fig. 3b) and imposes dynamic
stresses at the base of the overriding oceanic plate, leading to significant surface uplift (Fig. 3a). The
ascending plume experiences extensive decompression melting at the base of the overriding plate,
and due to the dynamic overpressure, spreads laterally, forming two branches that flow in opposite
directions (Fig. 3c). A large amount of plume material is eventually entrained towards the spreading
center, ponding underneath the ridge axis, and significantly affecting the ridge dynamics. The
entrainment of hot plume material promotes decompression melting (Figs. 3d, e) and increases the
temperature beneath the ridge (Fig. S2). Within the overlying lithosphere, the buoyant mantle plume
leads to stress localization and strongly weakens the oceanic plate (Figs. S1, S3). As the plume
eventually flows upward along the increasingly sloping base of the plate near the MOR, melting and
crust production occurs (Fig. S1), forming an oceanic plateau of thickened crust. In addition to this
gravitational force that guides plume material of the right branch ridge-ward, plate spreading drags
both branches in the opposite direction. Moreover, convective and tectonic stresses ("plume push"
and "ridge suction") affect both branches of the plume in a different way. As a consequence, the two
branches evolve asymmetrically: the right branch that flows towards the ridge axis is more vigorous
than the left branch, and the plume tail is also tilted towards the spreading center (Figs. 3c-e). For a
more detailed discussion of the underling controlling forces, see below.

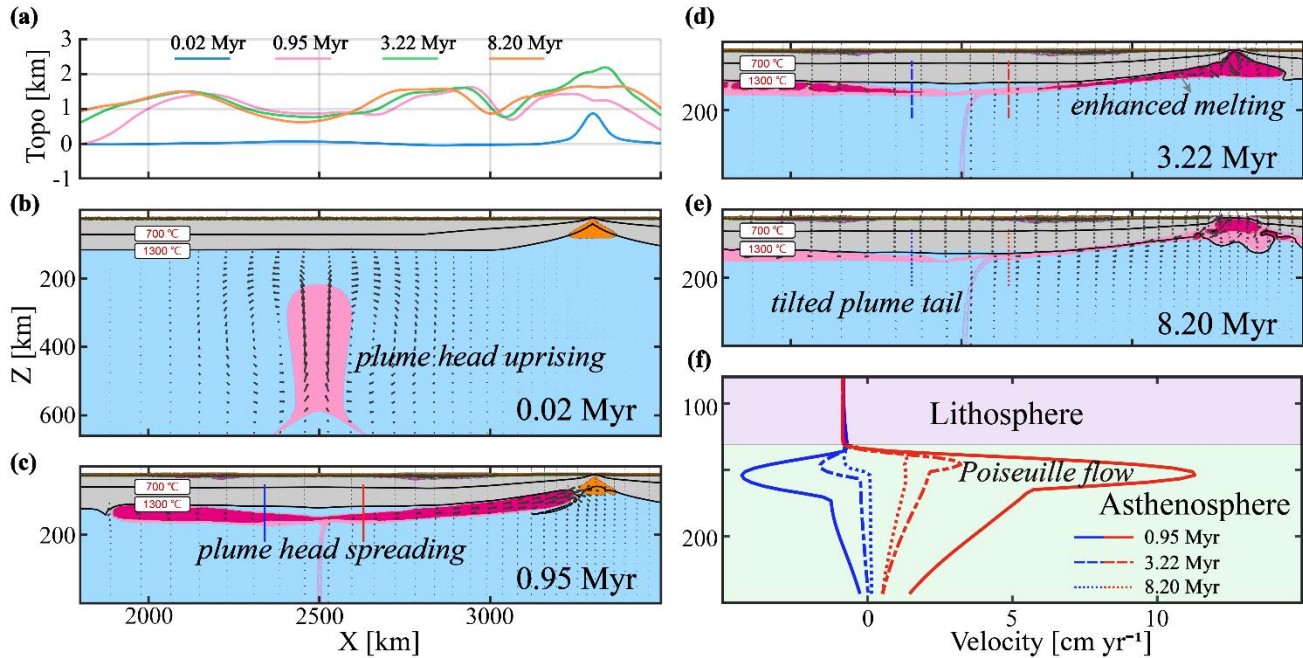

**Figure 3.** The evolution of the reference model M12 (see Table S1 in supplementary material) with

dominant ridge-ward plume flow. The main model parameters employed in this case are: half

spreading rate of 8 mm yr$^{-1}$, an initial plume head radius of 200 km, and an off-axis distance of 800

km. **(a)** surface topography over time along the flow path. **(b-e)** Snapshots of composition for the

reference model (M12). **(f)** Profiles of the horizontal velocity component over time at the sections as

indicated (color-coded) in panel (c-e).

The mantle flow horizontal velocity profiles (Fig. 3f) further demonstrate the dominance of

ridge-ward plume flow, showing that plume flow is faster towards the spreading ridge than away

from it. The velocity profiles elucidate dominant Poiseuille flow, with the maximum flow velocities

in the middle of the asthenospheric channel. Such velocity profiles are well consistent with

observations of seismic anisotropy at the Reunion plume (Barruol et al. 2019). The branches of the

spreading plume head move significantly faster than the overriding plate. Therefore, plate drag

actually slows down the spreading of the plume branches in this model case. Because of the

asymmetrical spreading of the plume head, the buoyancy flux carried by the right branch of the
plume (density anomaly multiplied by horizontal velocity from Figure 3f) is also much larger than
that carried by the left branch.

**3.2 Model evolution with plate-drag plume flow**
For models with dominant plume flow away from the ridge ("plate-drag flow"), the typical
model evolution is shown in Fig. 4. The controlling parameters of the representative model shown in
Figure 4 are the same as for the model shown in Figure 3, except for a smaller radius (100 km) and
faster spreading ridge (half spreading rate: 45 mm yr$^{-1}$). At first, the ascending plume head spreads
out similarly as in the case described above and interacts with the overriding oceanic lithosphere.
The largest surface uplift is sustained just above the plume head (Fig. 4a), slightly different from the
previous model in which the highest surface elevation is observed on both sides of the plume conduit
(Fig. 3a). Related to this spreading and uplift, divergent stresses are sustained in the overlying
lithosphere (Fig. S4), but no weakening or yielding occurs (Fig. S6). The plume head undergoes
significant decompression melting near the deflection point (Fig. 4c). However, thick and cold
lithosphere prevents magma from extracting (Fig. S4). As the plume cools, partially molten plume
gets solidified speedily (Figs. 4d-e and S5). In contrast to the reference model from section 3.1, this
model displays most plume material flowing away from the ridge, likely due to dominant plate drag
(Figs. 4c-e). Indeed, the left branch of the plume consistently displays larger buoyancy fluxes and
maximum velocities than the right side over time (Fig. 4f).

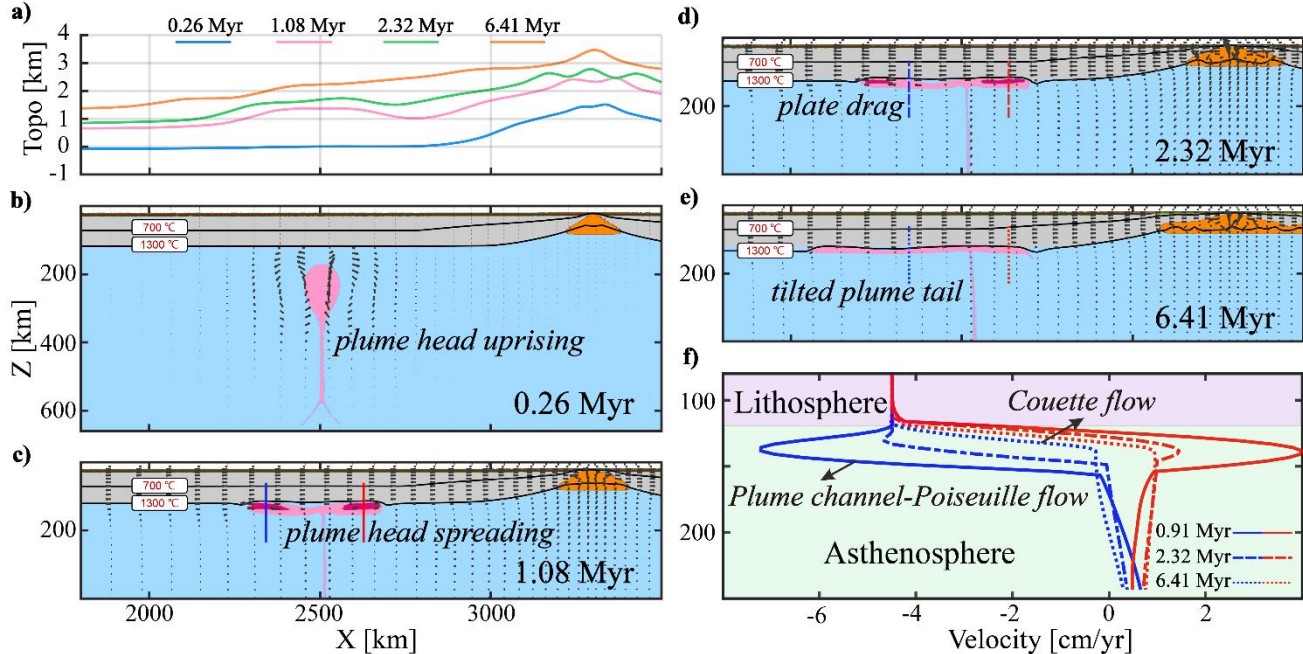

**Figure 4.** Same as Figure 3 for case M77 (i.e., the reference model for the plate-drag plume flow regime). The main model parameters employed in this case are: half spreading rate of 45 mm yr⁻¹, an initial plume head radius of 100 km, and an off-axis distance of 800 km.

The underlying mechanism for dominant plate-drag plume flow is the frictional shear force of the moving plate, which is further demonstrated by the plume flow velocity profiles (Fig. 4f). In the early plume head stage (~1.08 Myr), the plume spreads out faster than plate velocity, which is primarily driven by the overpressure of the ponding plume head at this stage. After a short amount of time (~2.32 Myr), however, plume spreading becomes significantly slower than plate velocity, and hence plate drag drives and controls the plume flow. Indeed, the flow mode in the asthenosphere rapidly shifts from Poiseuille flow (i.e., active plume flow) to Couette flow (i.e., passive plume flow) (Fig. 4f), indicating the increasing role of plate drag on plume flow, soon after an initial of plume-head spreading.

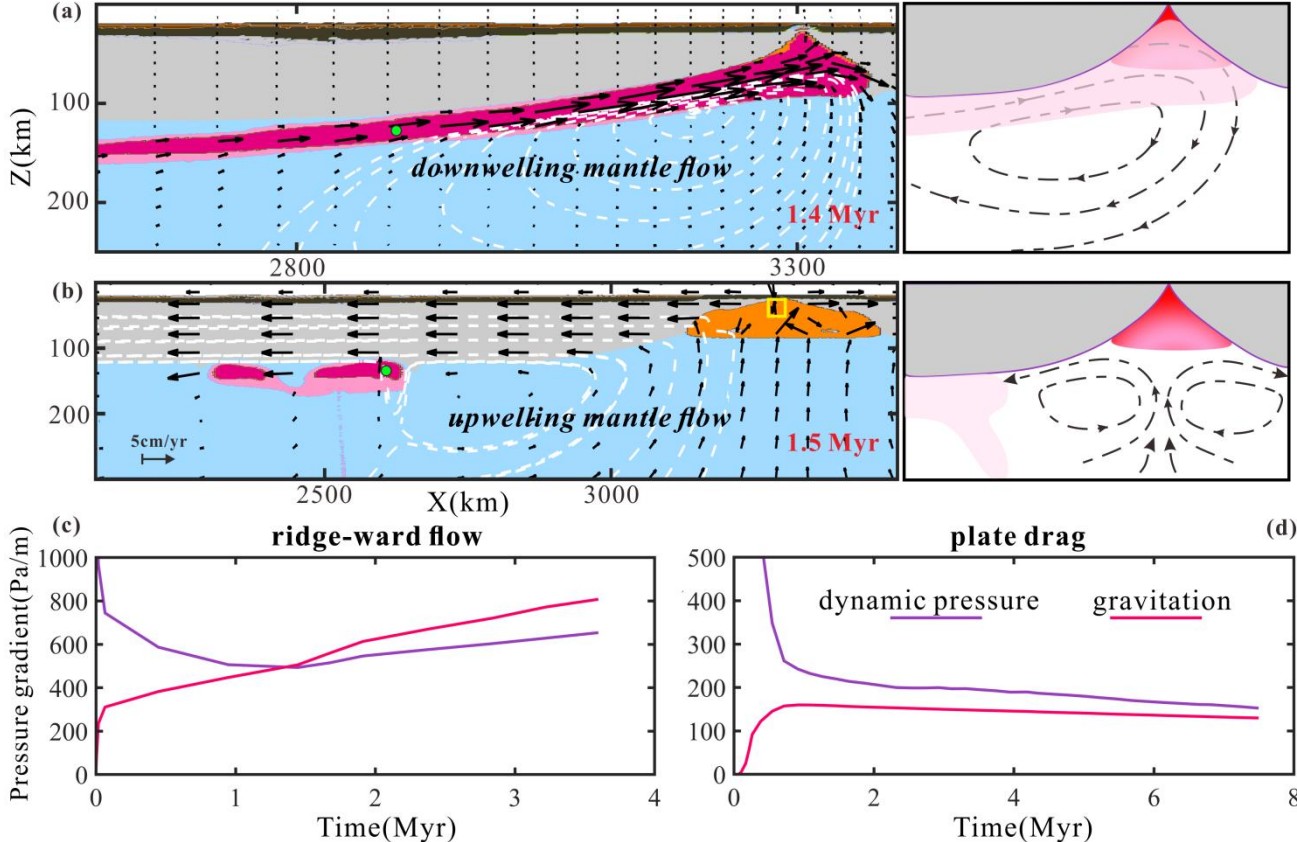

**Figure 5.** Comparsion between models with ridge-ward vs. plate-drag plume flow. **(a)** Ridge-ward

flow with downwelling beneath the MOR (results from case M12 as in Figure 3). White dashed lines

are streamlines; black arrows visualize the flow field. Schematic of flow in the sub-panel on the

right-hand side. **(b)** Plate-drag flow with upwelling mantle corner flow beneath the MOR (results

from case M77 as in Figure 4). **(c)** The dynamic pressure and gravitational gradient of plume marker

(i.e. green circle in (a)) over time. The yellow box in (b) marks the location for the computation of

average dynamic pressure at the ridge, needed for the calculation of the dynamic pressure gradient

(see text). **(d)** The dynamic pressure and gravitational gradient of plume marker (i.e. green circle in

(b)) over time.

## 3.3 Two modes of plume-lithosphere interaction

The dominant ridge-ward and dominant plate-drag plume flow regimes are two distinct modes of

plume-plate interaction. The differences between these two regimes are further demonstrated in
terms of mantle flow (Figs. 5a,b), driving forces (Figs. 5c,d).

In the ridge-ward dominated models, clockwise mantle develops from the plume to the spreading

ridge (Fig. 5a). Molten plume material flows to the spreading ridge and occupies the space
underneath the ridge axis, sustaining significant asymmetry of mid-ocean ridge melting (Conder et
al., 2002). As a consequence to the continuous supply of the plume material, downward mantle flow
forms beneath the ridge axis. This flow pattern dramatically differs from that shown in the plate-drag
dominated models, which show upward mantle flow underneath the ridge axis (Fig. 5b), as typical
for the flow beneath a MOR without the influence of a plume.

The distinct modes of plume-ridge interaction (ridge-ward vs. plate-drag flow) are controlled by

the competition of the tectonic (plate drag, ridge suction) and gravitational (plume buoyancy) driving
forces. On one hand, the moving plate drags sub-lithospheric plume material away from the ridge.
On the other hand, the mechanism of ridge-ward flow is twofold. First, the buoyant plume material
flows along the sloping base of the lithosphere towards the shallow ridge along the gravitational
gradient. Second, the plume is driven along the dynamic-pressure gradient from the pressure
maximum (e.g., where the plume sustains dynamic topograph) towards the pressure minumum
beneath the diverging ridge. These gravitational ( $G_{gv}$ ) and pressure-driven ( $G_{dp}$ ) gradients are
calculated by tracing plume markers (Figs. 5c,d) as follows:
$$G_{dp} = (P_{mk} - P_r)/L \qquad (12)$$

$$G_{gv} = (\rho_0 - \rho_{mk}) * g * k \qquad (13)$$

where $P_{mk}$ is the dynamic pressure of plume marker and $P_r$ is the averaged pressure in a 50 km box
at ridge center (Fig. 5b); $L$ is the horizontal distance from plume marker to ridge axis; $\rho_{mk}$ and $\rho_0$
are the plume marker density and initial density, respectively; $g$ is the gravitational acceleration; $k$ is
the local slope of the base of the lithosphere.
In the early stage of model evolution, the plume head's dynamic overpressure is dominant,
driving plume spreading in both directions (Fig. 5c), in particular in the direction of the low-pressure
ridge. However, this pressure gradient systematically diminishes over time as the plume (head)
spreads. Once the spreading plume approaches the ridge, the lithospheric slope increases. At some
point, the gravitational gradient exceeds the dynamic pressure gradient, taking over as the major
driving force of guiding plume material towards the ridge. Consequently, one of the essential
conditions for plume-ridge interaction is that the plume must be able to reach the critical zone near
the ridge, where the slope is sufficiently steep to take over for the ever diminishing pressure gradient.
This implies that the plume buoyancy must (1) overcome the shearing force of plate drag, and (2) the
pressure-gradient must be sustained long enough to reach the critical zone, in which the gravitational
gradient can take over. The (1) shearing force scales with the rate of ridge spreading, and the (2)
critical zone is more readily reached for high buoyancy fluxes at a given plume-ridge distance.


**3.4 Influence of model parameters**
We have systematically investigated the effect of the three main model parameters (i.e., the
spreading rate of the mid-ocean ridge, initial plume head radius and plume-ridge distance) on
plume-ridge interaction. We explored half spreading rates of the ridge of 8, 15, 30, and 45 mm yr$^{-1}$,

corresponding to ultra-slow, slow, medium, and fast-spreading mid-ocean ridges, respectively (Gerya, 2012). We varied initial plume head radii in the range of 100 km to 300 km. Further, we tested plume-ridge distance in the range of 600 to 1400 km.

**3.4.1 Plume head radius**

The size of the buoyant plume exerts an important control on plume-ridge interaction. Small plumes tend to be dragged away from the ridge, with typically larger lateral fluxes of the left branch than the right branch of the spreading plume (Figs. 6a,b). The buoyancy flux in each branch is calculated by multiplying the velocity of the markers in plume pipe (Figs. 6d-f) by the density. The dynamic pressure decreases with decreasing plume size (Fig. S8a), and the pressures gradient is thus not strong enough for small plumes to reach the ridge. Plate shearing dominates plume flow soon after plume head spreading, and the moving plate then drags plume head material, leaving a tilted plume tail (Fig. 6d). In contrast, with larger initial plume head radius or buoyancy flux, the ponding plume spreads more vigorously (Fig. 6c) and sustains much higher overpressures at the base of the plate (Fig. S8a). This vigorous spreading can overcome plate drag to drive Poiseuille flow in both directions. Once the right plume branch approaches the spreading center, it is attracted and further accelerated by ridge suction. The plume tail is also markedly tilted towards the ridge axis due to asymmetric spreading (Fig. 6f). The larger the plume is, the more plume material gets entrained by the spreading center.

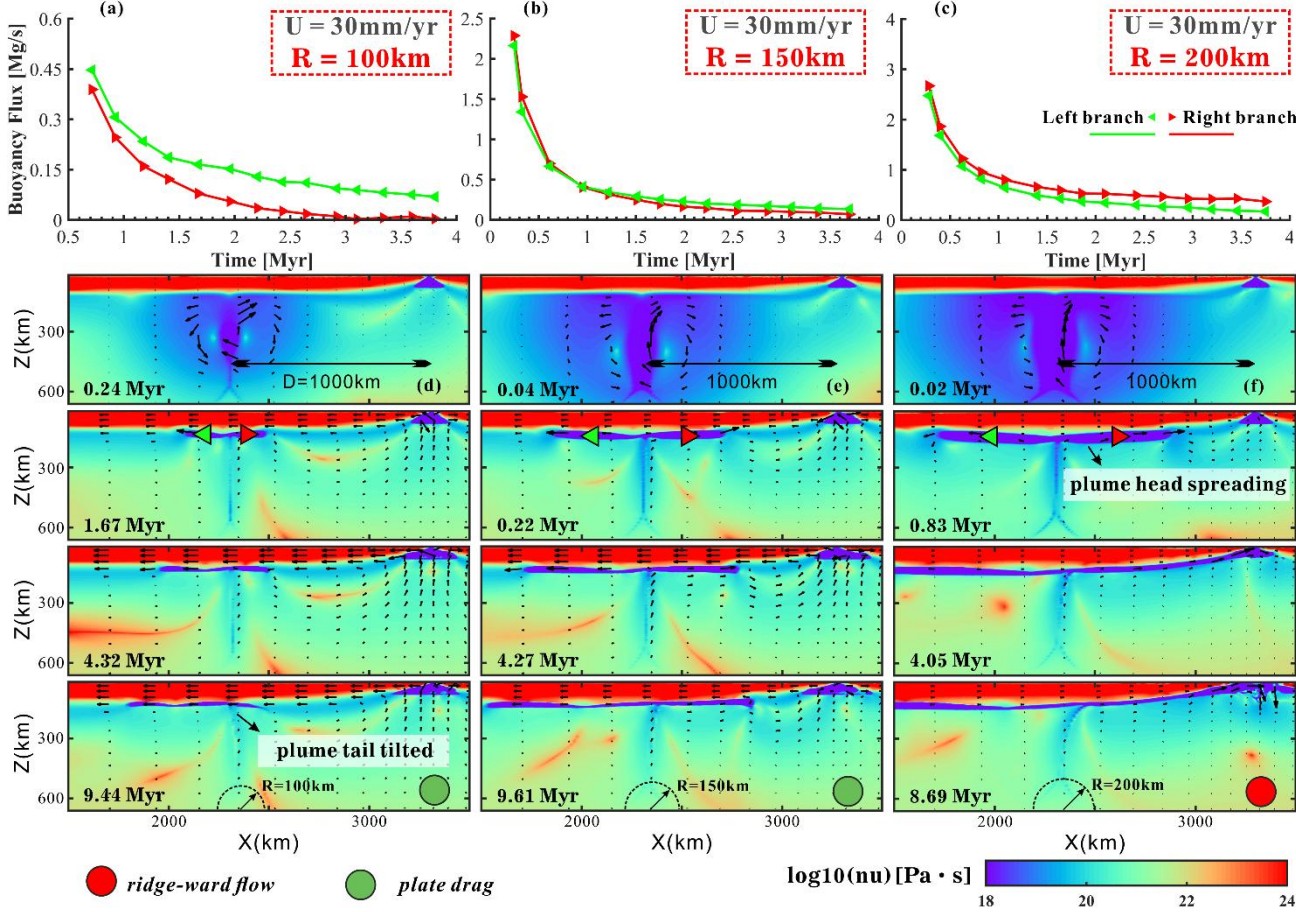

**Figure 6** Models varying initial plume head radii (model M53, M58, and M63, Table S1 in supplementary material) shown by buoyancy flux and viscosity. **(a-c)** Buoyancy flux in spreading plume branches over time. Green and red triangles are markers used for buoyancy flux calculation. **(d-f)** Viscosity snapshots of models with different plume head radii. Models with green circle represent plate-drag plume flow and ridge-ward plume flow in red.

### 3.4.2 Plume-ridge distance

Plume-ridge distance also controls the regime of plume-ridge interaction. A plume at large distances spreads similarly as a plume at a small distance, but is less likely to get affected by ridge suction (Figs. 7e,f). The pressure gradient between the plume and ridge drives the ridge-ward plume flow. However, the larger the plume-ridge distance, the smaller the pressure gradient would be (Fig.

S8b), resulting in a lower buoyancy flux across the plume pipe (Figs. 7a-c). In the cases of distant
plumes, the spreading of the plume head is strongly affected by plate drag (Figs. 7b, c). On the other
hand, the difficulty in sustaining ridge-ward plume flow may also link to the heat transfer between
the cold plate and the hot plume rocks. With gradually cooling from upper plate by heat conduction
and diffusion, the viscosity of plume increases as it cools. Such increasing viscosity slows the plume
down, stopping the ridge-ward plume flow eventually (Figs. 7e, f). Hence, for cases with large
plume-ridge distances and hence travel times, the ponding plume head cools and is ultimately carried
away by the moving plate.

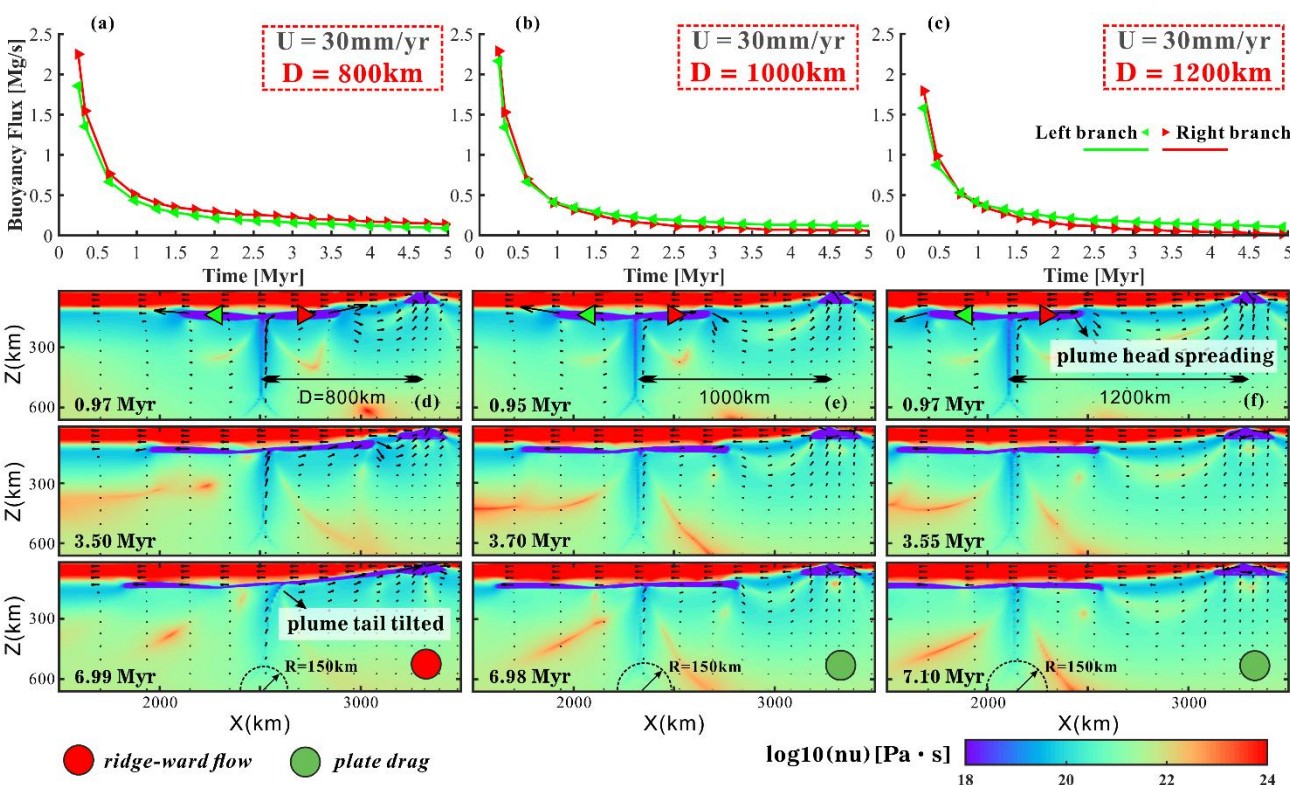


**Figure 7.** Models varying plume-ridge distances (model M57-M59, Table S1 in supplementary
material) shown by buoyancy flux and viscosity. **(a-c)** Buoyancy flux in spreading plume branches
over time. Green and red triangles are markers used for buoyancy flux calculation. **(d-f)** Viscosity
snapshots of models with different plume-ridge distances. Models with green circle represent
plate-drag plume flow and ridge-ward plume flow in red.

### 3.4.3 Half spreading rate of ridge

Another parameter that is worth investigating is the spreading rate of the ridge. The modeling
results indicate that fast-spreading ridges promote plume flow away from the ridge due to the friction
(Figs.8 and 9a). With increasing spreading rate, the effect of plate shearing on plume-lithosphere
interaction increases, as quantified by the spreading fraction. The spreading fraction $\gamma$ (Eq.(14)) is
defined here as the ratio of ridge-ward vs. plate-drag plume volume fluxes. We integrated the
ridge-ward plume volume flux (right branch), $V_{rw}$, and plate-drag plume volume flux (left branch),
$V_{tw}$. $V_p$ is the total plume volume flux in the model. Ridge-ward plume spreading is dominant for
positive $\gamma$; plate-drag plume spreading is dominant for negative $\gamma$.

$$\gamma = (V_{rw} - V_{tw})/V_p \qquad (14)$$

In the early stage (~1 Myr), pressure-driven flow dominates in all models and spreading
fractions are positive, mainly driven by the expansion of the overpressured plume heads along the
pressure gradient. After a certain time, the spreading fractions decrease dramatically with the decay
of the mantle plume activity, representing the transition from the ridge-ward to the plate-drag regime
in some cases. The characteristic spreading fractions after 8 Myr model time as a function of our
model parameters are shown in Fig. 8. This compilation of our results reveals that the dominance of
ridge-ward flow decreases with increasing spreading rate and off-axis distance, but significantly
increases with plume size. For models with fast-spreading ridges, the parameter range of plate-drag
flow dominated models is expanded, indicating the critical role of plate drag in restricting ridge-ward
flow and plume-ridge interaction.

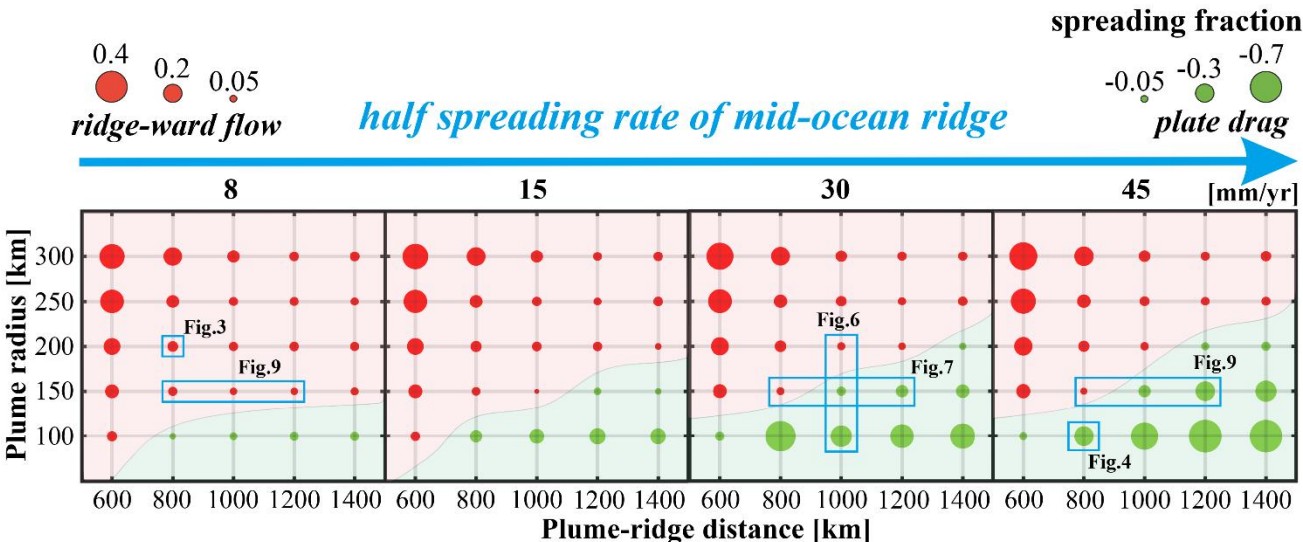

**Figure 8.** Parameter regime diagram of the contrasting modes of plume-ridge interaction. Spreading fractions $\gamma$ (Eq. (14)) at ~8 Myr model time. Each of the circles represents one of the numerical experiments, and sizes refer to $\gamma$. Circles in red and green represent models with dominant ridge-ward plume flow and plate drag, respectively.

The transition from ridge-ward (positive $\gamma$) to plate-drag (negative $\gamma$) flow in some of our cases is mainly determined by the competition between the effects of pressure-driven plume head spreading and plate shearing. The overpressure in the plume head drives plume materials towards the lower pressure spreading center, while the moving plate shears plume away. Hence, we quantify the shear force of the overriding oceanic plate on the plume head using an integral approach:

$$F_s = \int \sigma_{xz}\, dA \quad (15)$$

Equation (15) is employed to calculate the shear force, where $F_s$ is the total shear force the spreading oceanic plate exerts on the uppermost part of the plume. $\sigma_{xz}$ is the shear stress on each mantle plume grid cell, $A$ refers to the area of each grid cell. The pressure gradients, both

gravitational and dynamic pressure, are calculated by tracing the plume markers according to
equations (12-13). As the plume material rises to the base of the lithosphere, the shear force exerted
by the plate increases over time. We find that the integrated shear force between the spreading plate
and the plume increases significantly as half spreading rate increases (Fig. 9c).
Conversely, ridge spreading rates control gravitational and pressure-driven plume driving forces
(Fig. 9d). Increasing the spreading rate of the ridge implies a smaller dynamic pressure gradient,
because the pressure gradient is related to the plate thickness difference at the ridge and plume,
which is dependent on the spreading rate. A fast-spreading ridge also implies a smaller gravitational
gradient, because it leaves a more shallowly-dipping lithospheric base. Thus, relatively strong plate
shearing combined with relatively small pressure and gravitational gradients tend to advance
plate-drag plume flow for high spreading rates.

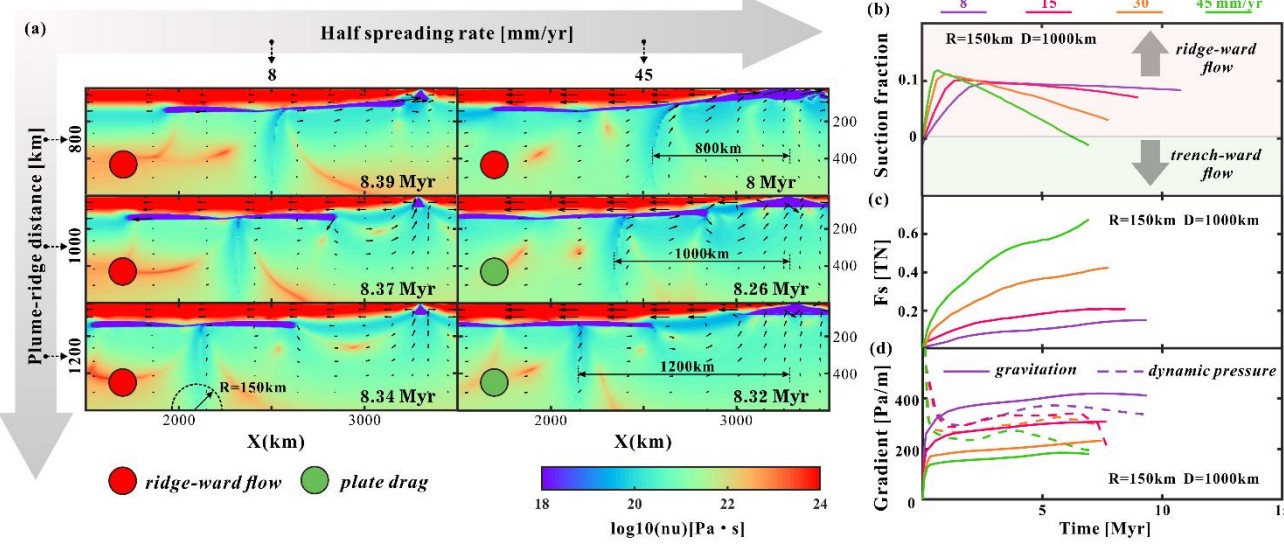


**Figure 9.** Model results influenced by different half spreading rates. **(a)** Effect of spreading rate on
ridge-ward flow vs. plate-drag flow. Viscosity snapshots are shown (model M7-M9, M82-M84,
Table S1 in supplementary material). Fast-spreading ridge promotes plume material dragged. Models
with green circle represent plate-drag plume flow and ridge-ward plume flow in red. **(b)** Dynamic
evolutions of ridge-ward and plate-drag plume flow, revealed by defined ridge spreading fraction
(eq.14). **(c)** Shear force (*Fs*) between moving plate and plume material under different spreading
rates. **(d)** Pressure gradient between plume head and ridge center in different half spreading rate
models. The solid and dash lines are the plume gravitation and dynamic pressure gradient,
respectively.


**4  Discussion**
Natural observations show that there are only very few hotspots indicative of ridge-ward plume
flow close to the East Pacific Rise (EPR) (Fig. 10a), in contrast to many such hotspots in the Atlantic
and Indian oceans. A previous study (Jellinek et al., 2003) proposed that fast-spreading ridges such
as the EPR efficiently convey any surrounding plumes into the spreading center from the deep
mantle (Fig. 1c), which leads to fewer hotspots nearby fast-spreading ridges. However, based on our
modeling results, fast-spreading ridges tend to promote plate-drag flow of the spreading plume
material, providing an alternative explanation to the relatively absence of hotspots along the EPR.
We discuss the viability of this potential explanation by comparing with geological and geophysical
observations (Fig. 10).
Firstly, the plate drag effect of fast-spreading ridges on plumes is evidenced by geophysical
observations. We locate the positions of the mantle plumes at the core-mantle boundary (CMB) and
the associated hot spots on the surface based on global seismic tomography (Jackson et al., 2021;
Koppers et al., 2021). A lateral offset between the deep and surface positions of plumes is a common
feature, indicating the deflection of plumes due to mantle flow. Specifically, a large portion of
plumes located in the Atlantic are tilted towards the mid-ocean ridge. However, only very few
plumes in the Pacific are tilted towards the mid-ocean ridge; indeed, the majority of plumes are tilted
away from the ridges, indicating the significant effect of plate drag on plumes beneath fast plates.
Such observations are consistent with the predictions of our models with dominant plate-drag plume
spreading.

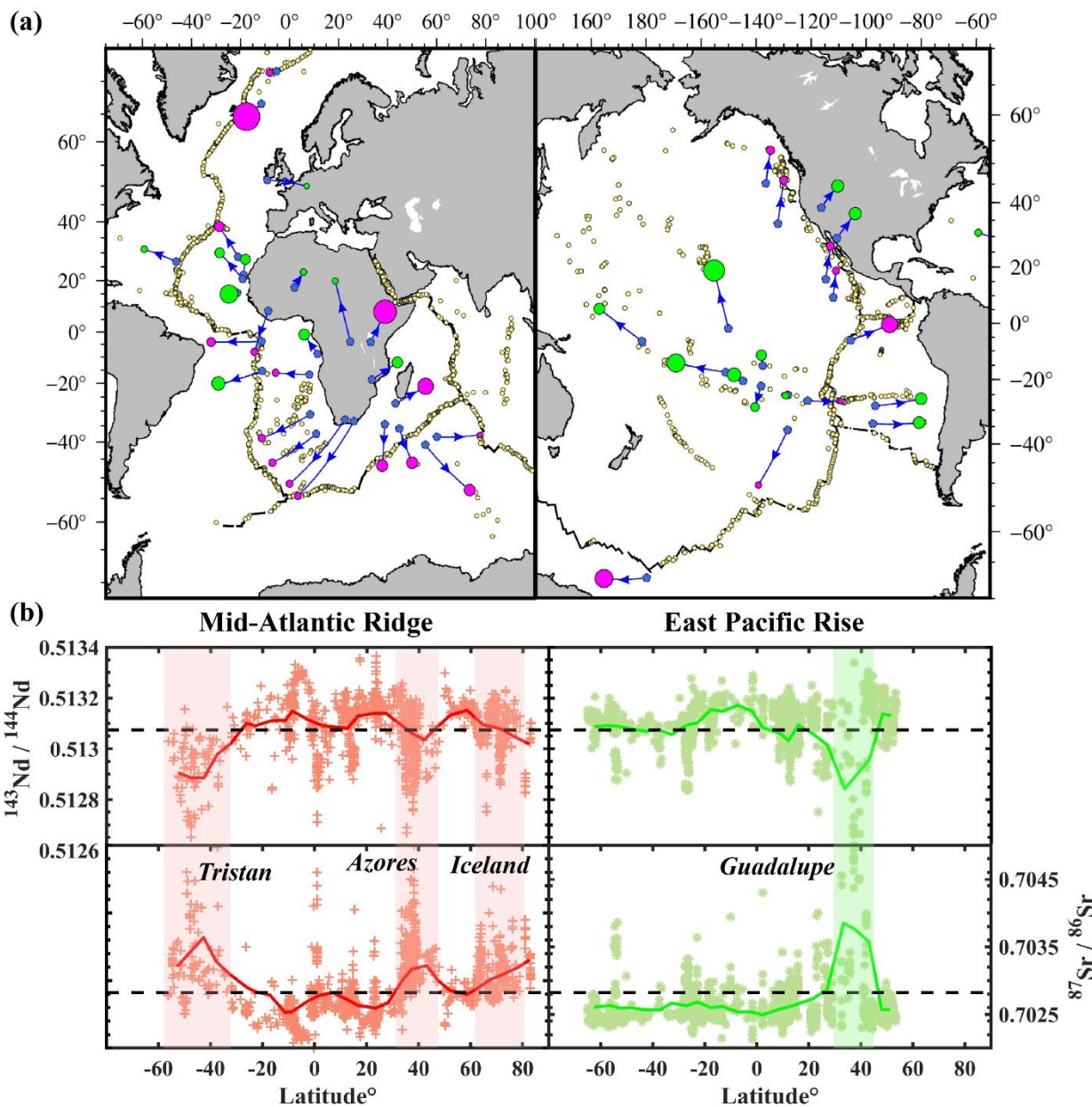


**Figure 10.** A compilation of hotspots along with spreading ridges in the Atlantic and the Pacific. **(a)**
Distribution of surface hotspots (circles) together with depth-projected source locations at CMB
(blue dots) of the plumes based on (Jackson et al., 2021). Plumes in magenta circles are mantle
plumes interacted with ridges (Ito et al., 2003), and plumes not interacted with ridges are shown as
green circles, whose size refers to the plume buoyancy flux (Hoggard et al., 2020). Yellow dots are
MORB samples mapped in (b). **(b)** Plot of radioactive isotopes ratios along ridge MORB samples.
The data are downloaded from the PetDB Database (http://portal.earthchem.org/). The colored
symbols refer to samples in different mid-ocean ridge. Main hotspots influencing MORBs are
labeled with shaded bands. The black dash lines are the mean MORB isotopes ratio from Gale (2013).
Red and green lines are the mean ratios of the samples in Mid-Atlantic ridge and EPR, respectively.

Geochemical studies suggest that mantle plumes are enriched in light rare earth elements

(LREEs) and radiogenic isotopes of Sr and Pb but depleted in Nd isotopes. These geochemical
anomalies are evident in MORB at the sites of active plume-ridge interaction (Cushman et al., 2004;
Douglass and Schilling, 1999; Yang et al. 2017). We find that MORB sampled along both the
Mid-Atlantic ridge and the EPR indeed display geochemical anomalies (Fig. 10b), indicating
ridge-ward flow of plume material at specific locations. However, the Mid-Atlantic MORB dataset is
slightly more heterogeneous than the East Pacific Rise in terms of geochemical isotopes. The EPR is
basically characterized as normal oceanic basalt, along which only very few regions show
composition associated with nearby plumes. This contradicts the view (Jellinek et al., 2003) that
mantle plumes are fully entrained into the central MOR melting zone at fast-spreading ridges.

Based on our modeling results, initial plume head radius and plume-ridge distance also control

the mode of plume-ridge interaction. However, there is only a small difference in terms of the
fraction of interacting vs. non-interacting plumes for different buoyancy fluxes $B$: a small majority of
major plumes (5 of 8 with $B > 1.6$ Mg/s) vs. a small minority of small-to-intermediate plumes (11 of
25 for $B < 1.6$ Mg/s) display interaction with the ridge (Fig. 11a). The underlying cause for this
observation remains unclear, but may be related to the distribution of large plumes globally with
many of them being located very far from MORs. Also note that our 2D models are limited in that
plume material cannot spread in the out-of-plane direction, hence somewhat exaggerating the effects
of buoyancy flux. In any case, the distribution of observed plume buoyancy fluxes (Hoggard et al.,
2020) varies little across different oceans (Fig. 11a). Therefore, the effects of plume size are not a
good candidate to explain the notable difference between the Atlantic and Pacific in terms of
plume-ridge interaction mode.

On the other hand, compared with the Atlantic and Indian Oceans, Pacific plumes are located

significantly further from the mid-ocean ridge (Fig. 11b). Plume-ridge distances in the Pacific are
mostly >2000 km, which exceeds the maximum plume-ridge interaction distance of 1400 km
(Schilling, 1991). Most plumes in the Pacific exhibit the typical signatures of plume flow away from
the ridge, such as parabolic swell shapes (e.g., Society, Marquesas and Hawaii plumes; Ballmer et al.,
2013a; Ballmer et al., 2015; Cheng et al., 2015; Wolfe et al., 2009), and linear volcanic chains (Buff
et al., 2021; Clouard and Bonneville, 2005; Jackson et al., 2010). Age-progressive hotspots trails
indicate an absence of dominant ridge-ward flow. By contrast, most plumes in the Atlantic have been
close to the ridge since the opening of the ocean. These mantle plumes (e.g., Discovery, Iceland,
Tristan-Gough; O'Connor et al., 2012) did not move much since the breakup of the Atlantic. One
factor may be that the underlying plume generation zone (i.e., the edge of the African LLSVP) round
largely parallel to the Mid-Atlantic Ridge (Fig. 1) (Torsvik et al., 2006). In this case, plume-ridge
distance may play a critical role in the plume-ridge interaction, and could explain the striking

difference between the Pacific and Atlantic in terms of the number of plume-ridge interacting vs.

non-interacting systems. In addition, the rapid movement of the Pacific plate tends to inhibit

ridge-ward plume flow at a given plume-ridge distance. The distribution of interacting (stars) vs

non-interacting systems in Figure 11b is almost exactly as predicted by our models for the coupled

effects of plume-ridge distance and plate velocity. For example, we note that fast-spreading ridges

can still interact with adjacent plumes under the appropriate conditions. In the case of very short

plume-ridge distances, there is good evidence of plume-ridge interaction in the southern Pacific

ocean (e.g., Louisville plume; Conder et al., 2002; Toomey et al., 2002; Vlastélic and Dosso, 2005).

Based on a series of numerical modeling as well as geological and geophysical observations, we

conclude that mantle plumes in the Pacific are more likely to spread away from the ridge and into the

direction of plate motion than in the Atlantic and Indian Oceans. The tendency of fast plate velocities

to promote plume spreading away from the MOR through viscous drag may depend, however, on the

details of lithosphere-asthenosphere rheological coupling such as the presence of a weak decoupling

(e.g., melt) layer (Rychert et al., 2020). Further studies of plume spreading and plume-ridge

interaction are needed to shed light on the coupling of the plate-mantle system.

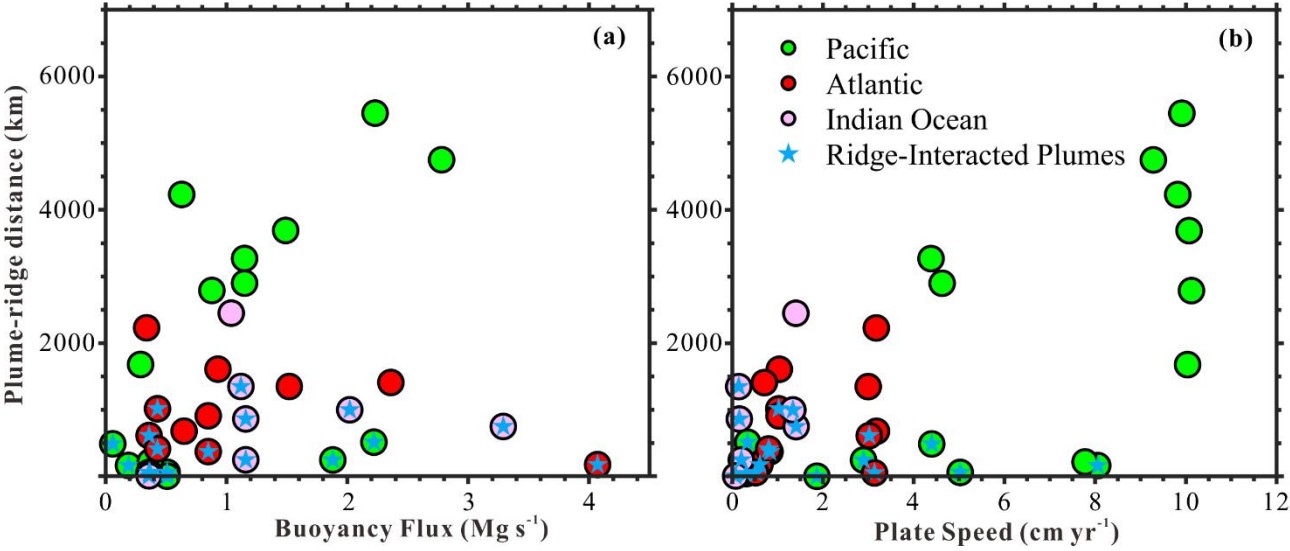

**Figure 11.** Buoyancy flux, plate speed and plume-ridge distance of mantle plumes in different oceans. Mantle plumes in the Pacific, Atlantic and Indian Ocean are shown in green, red and pink circles, respectively. Blue stars marked the ridge-interacted plumes according to Ito et al. (2003). **(a)** Plot of plume-ridge distance and plume buoyancy flux. Data are from Hoggard et al. (2020). **(b)** Plot of plume-ridge distance and plate speed at the location of plumes. Plume-ridge distance come from GPlates (Müller et al., 2016; Whittaker et al., 2015), and plate speed data come from Becker et al. (2015)

## 5    Conclusion

In this study, we explore the evolution of plume-ridge interaction with 2D thermomechanical numerical models. Based on model results, we find that:

(1) Plume-ridge interaction is mainly governed by the competition between the effects of plume spreading (overpressure in the plume-head stage), upward gravitationally-driven flow of the plume along the base of the sloping lithosphere and plate shearing. These driving forces are controlled by plume size, plume-ridge distance and the spreading rate of the mid-ocean ridge.

(2) MOR spreading does not only draw upwelling plumes into the spreading center, but also tends to drag mantle plumes away from the ridge. Plume flow away from the ridge is favored by small and/or distant plumes as well as fast spreading rates, whereas plume flow towards the ridge is promoted by large and/or nearby plumes, as well as slow spreading rates.

(3) Considering the high plate velocity and typically large plume-ridge distances, mantle plumes in the Pacific are more likely to be dragged away from the EPR than being drawn towards the ridge center.

543

544

**Code availability**

The source numerical modeling code in this study is available from the corresponding author upon reasonable request.

**Data availability**

The data that support the findings of this study are available from the corresponding author upon reasonable request.

**Author contribution**

Fengping Pang performed all numerical models, interpreted results and wrote the manuscript. Jie Liao proposed the study, modify the code and contributed to rewriting and scientific discussion. Maxim D. Ballmer contributed with significant help in rewriting and scientific discussion. Lun Li participated in discussion and interpretations. All authors have read and edited draft versions of the paper and have approved the final version.

**Competing interest**

The authors declare that they have no conflict of interest.

**Acknowledgement**

This research is financially supported by NSFC projects (U1901214, 41974104, 91855208) and Guangdong project 2017ZT07Z066. We are grateful to Prof. Taras Gerya for his long-lasting guidance on our geodynamical modeling. We gratefully acknowledge Hongjian Fang for insightful

discussions. Numerical simulations were performed on the clusters of National Supercomputer
Center in Guangzhou (Tianhe-II).

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
