# Peer review of "Plume-ridge interactions: Ridge-ward versus plate-drag plume flow"

_Solid Earth, 2022_

## Author Comment (AC1)

Dear referee,

We are glad to receive the review report and would like to express our sincere thanks to you and the editors. Without the constructive comments from you, the quality of this manuscript cannot be significantly improved. All the comments and suggestions have been carefully considered to revise the manuscript. Detailed reply to all comments and the associate manuscript modifications are given below.

**Reply on RC1**

1. The language of the manuscript needs to be fully checked and revised by a professional editing service or a native speaker

*Reply: We appreciate your advice and have revised the manuscript carefully to improve the language, with the help of all the co-authors.*

2. In this study, the relation between spreading rate and age of oceanic lithosphere is ignored. Usually, higher spreading rates create younger lithospheres at a constant distance. In this study, the authors assumed that the lithospheric age is constant near the side boundaries (50 Myr). As a result, by imposing some higher velocities near the side boundaries to simulate higher spreading rates, the lithosphere becomes under extension and since the ridge is the weakest point in the system the width of ridge changes (it is clearly seen in e.g., Figs 3e,4e and 5a-b); higher rates lead to wider ridges. Could the authors explain to what extend is this assumption realistic? I think the formation of cracks in the lithosphere is the consequence of this assumption

*Reply: We appreciate this comment which triggers us a lot of thinking. We further checked our model results, especially the stress state in the whole lithosphere. The result shows that the lithosphere seems under extension owing to the high internal velocities (Figs. S1, S4). However, there is no similar stress localization in the other side of lithosphere that without plume-lithosphere interaction. Actually, the "tension cracks" used in the main text may be inaccurate. There is no weakening or plastic deformation in the lithosphere, and there are no normal faults near the surface in the models. We*

*describe the stress distribution in the lithosphere to highlight the stress localization, which occurs in the lithosphere where plume flows to the ridge. As a result, we revised and simplified the description of the "cracks" in the main text (lines 212-213 and lines 249-250).*

[Figure]

**Figure S1.** Reference model (M12, see Table S1, same as Fig. 3) evolution of ridge-ward plume flow shown by (a) crust and sediment thickness, (b) normal stress. The mantle plume weakens the overlying oceanic plate and changes the stress state of the overlying oceanic plate. Molten plume material beneath the lithosphere is extracted to the crust.

[Figure]

**Figure S4.** Reference model (M77, see Table S1, same as Fig. 4) evolution of trench-ward plume flow shown by (a) crust and sediment thickness, (b) normal stress. The mantle plume weakens the overlying oceanic plate and changes the stress state of the overlying oceanic plate. Molten plume material beneath the lithosphere is extracted to the crust.

3. In the abstract it is written "plume migration driven by plate drag is promoted by fast-ridge spreading rate." This is true only if the plume radii are small. For large plumes the rate of spreading is irrelevant (Fig. 6). This should be mentioned here and also in discussion and conclusions.

*Reply: We agree with you and rephrased this sentence (lines 19-20). Indeed, plate dragging is most significant when the plume buoyancy is relatively small. When the plume is buoyant enough, plate drag plays a minor role than the plume self-spreading*

*on plume-ridge interaction. We have revised this part and the discussion section in the manuscript.*

4. Usually decomposition melting of plume head causes the formation of a plateau above the plume head. Where do plateaus form in the models? I suggest that the authors add information about where plateaus form and how thick the crust is to the manuscript. The temporal evolution of plateaus is also interesting to be investigated.

*Reply: We appreciate this suggestion and add figures to present the temporal evolution of the extracted melt, displaying the crust thickness in the model over time (see figure S1, S4 in the supplementary material). Mantle plumes melt beneath the lithosphere and are then extracted into the oceanic crust, converting into basalt to form a thickened oceanic crust (Specific mechanism described in method section). In the model, oceanic plateaus (thickened crust) are formed directly above the spreading plume head. To describe plateau formation in our models, we amended the text in lines 213-215 and 251-252.*

5. Lines 20-22: "Our results highlight fast-spreading ridges exert strong plate dragging force, rather than suction on plume motion, which sheds new light on the natural observations of plume absence along the fast-spreading ridges, such as the East Pacific Rises." As I indicated above this is true only if plume radii are smaller than 250 km (based on Fig. 6). This conclusion implies that plumes in the Pacific are smaller than those in Atlantic. Are there any observations supporting this? I'm interested in a discussion about this issue in the paper.

*Reply: Thank you for your comments. We revised this sentence (lines 24-25). Observations show approximate plume buoyancy flux distributions in the Pacific and Atlantic (Figure 1 below). There is no obvious correlation between the distribution of buoyancy of mantle plumes in different oceans with their spreading rate. We conclude that plume size is not the deciding factor to explain the difference between the Pacific and Atlantic in terms of plume-ridge interaction mode. We add a figure and discussion about this in the discussion section (lines 470-481). Please also see our reply to*

*comment 6.*

[Figure]

Figure 1. Buoyancy flux, plate speed and plume-ridge distance of mantle plumes in different oceans. (a) Histogram of plume buoyancy flux distributed in the Pacific and Atlantic. The guassian distribution curves are shown in light blue and red lines, respectively. (b) The plot of plate speed at each plume and their off-axis distance. Blue stars mark the plumes shown to be interacted with the nearby ridges.

6. Looking at distribution of plumes and their sizes in Fig.9, one cannot see any correlation between plume size with plate drag and ridge suction. Can authors comment on that? Besides, in conclusion it is written: "The plume size, that is, the plume buoyancy flux, may play a critical role in controlling the connection between the two units, compared with distance and spreading rate." Why does plume size play important role compared to the two other factors? This is not discussed in the main text

**Reply:** *Thanks for your comments. We suggest that all three factors play an important role (Fig. 8). The predicted effects of plate velocity and plume-ridge distance are fully consistent with observations (Fig. 11b). The effects of plume buoyancy flux are less obvious when compared to observations (Fig. 11a). For discussion, see lines 470-481. We also reconsidered the importance of different influence factors and rephrased the sentence in the conclusion.*

[Figure]

**Figure 11.** Buoyancy flux, plate speed and plume-ridge distance of mantle plumes in different oceans. Mantle plumes in the Pacific, Atlantic and Indian Ocean are shown in green, red and pink circles, respectively. Blue stars marked the ridge-interacted plumes according to Ito et al. (2003). **(a)** Plot of plume-ridge distance and plume buoyancy flux. Data are from Hoggard et al. (2020). **(b)** Plot of plume-ridge distance and plate speed at the location of plumes. Plume-ridge distance come from GPlates (Müller et al., 2016; Whittaker et al., 2015), and plate speed data come from Becker et al. (2015)

7. Line 413- 415: "Based on a series of numerical modeling as well as geological and geophysical observations, we predict that mantle plumes in the Pacific Ocean are more likely to be dragged away by the spreading ridge." The authors emphasize on the fast spreading rate of Pacific ocean as a main factor for dragging plumes away from the ridges. I think the plume-ridge distance may be a main factor in this case; most of plume tails (shown as blue dots in Fig. 9a) in the Pacific Ocean are located away from the ridges.

*Reply*: *We appreciate this comment and agree with the reviewer. We reorganized the manuscript discussion (mostly in the final paragraph of section 4) accordingly.*

8. Line 145: Temperature of 2513 K is very high for temperature at 660 km. Considering adiabatic temperature gradient of 0.5 K km-1 and temperature of 1573 K at the base of lithosphere, temperature at the bottom of model should be ~1873 K.

*Reply: The model sizes in our study are set as 6600(width) and 1200(depth). The temperature of 2513K refers to the temperature at the bottom of model, that is, the temperature at 1200 km. Based on adiabatic temperature gradient of 0.5 K km$^{-1}$, the temperature at 660 km in all models is set initial to 1573 K + (660-120)km\*0.5K/km =1843K.*

9. Fig. 5: I expect the heat flux and melt are initially maximum in the area above the plume head. Then due to underplating of plume and its flowing towards ridge, the location of maximum heat flux and melt changes in time. That would be worth to show the evolution of heat flux and melt in time (similar to what is shown for surface topography in Fig. 3 and 4). For (5e-f): I suggest to show the results of plate drag model from ridge to some distances away from it, similar to what is shown in (c) and (d). I suspect that in plate drag due to imposing higher extension rate, the whole lithosphere is experiencing cracks and becomes extremely weak. Figure 5 shows the results in the early stage of deformation. Can authors provide a figure showing results at later stages?

*Reply: We appreciate this suggestion and add figures to present the temporal evolution of the extracted melt (see figure S1, S4 in the supplementary material). The maximum extracted melt above the plume head, shown as crust thickness in Figure S1, S4, change with flowing of the plume, which is consistent with the topography evolution in Figure 3a and Figure 4a. Besides, considering that the heat flux is not that important to our model result (suggested by another referee), we removed the description of heat flux evolution but reserve the change of extracted melt over time.*

*Secondly, we also present the lithosphere stress in a wider perspective (Figures. S1, S4). We agree with you. The whole lithosphere is under extension sightly because of the imposed extension rate. The presence of buoyant mantle plume leads to the stress localization within the above lithosphere. The normal stress in blue represents*

*extension, while the compression region is plotted in red. Actually, since plastic deformation does not occur in a widespread manner in our models, it indeed does not seem to be appropriate to describe such localized stress areas as "tension cracks". Please also see reply to comment 2. As a result, we removed the description of "cracks" in the main text (lines 212-213 and lines 249-250).*

**Other comments**

Line 36: What is Amsterdam?

**Reply:** *The Amsterdam here means the Amsterdam-Saint Paul mantle plume. We have made revision in the main text (line 41).*

Lines 136-138: It is not clear what this sentence mean. Please modify this sentence.

**Reply:** *We rephrased this sentence (lines 170-172) as "To reproduce the oceanic lithosphere, we choose a typical layered model, where the crust is composed of a water level (2 km), a sediment layer (1.5 km), a basalt layer (7.5 km)."*

Lines 147-148: This is not consistence with cooling half space. The temperature of the oceanic lithosphere tends to change linearly with depth when lithosphere is very old (older than ~80 Myr).

**Reply:** *Thanks for your comments. Indeed, the initial temperature distribution of the oceanic plate is prescribed by the half-space cooling model and thermal equilibrium structure. The half-space cooling model is used to describe the oceanic plate younger than 50Myr, and the thermal equilibrium structure is used to describe older oceanic parts. The thickness of the half-space cooling part is defined by the thermal structure. We set up the models in this way because we consider that the theoretical half-space cooling model has a good match with some geophysical observations when the plate is young, but the fit becomes poor when the age is greater than 60/70 Ma (Turcotte and Schubert, 2014; Stein and Stein, 1994). Therefore, we set a half-space cooling model with a maximum age at 50Ma, and the thermal equilibrium thickness of the older*

*lithosphere is constant (i.e., ~100 km; corresponding to a thermal age of 50 Ma). We have made further description of the model initial setup in the main text.*

Line 153: "An additional velocity is imposed on both sides of the ridge to represent the half spreading rate. "Are they internal boundaries? Please explain more about it here; where are they and until which depth they extend.

*Reply: Yes. The velocity boundaries are internal boundaries, which are imposed on 500 km from each side of boundaries in the lithosphere (i.e., from 20 km to 120 km in depth). We made revision in the main text (lines 97-98; lines 187-189).*

Line 181: "The mantle flow vertical velocity profiles" It is a bit confusing. The profiles shown in Fig. 3f are the horizontal component of mantle velocities along two vertical profiles. Please rephrase this part and also explain the depths which were selected for these profiles. Are they from the surface till ~250 km depth?

*Reply: Thanks for your suggestions. The profiles shown in Fig. 3f and Fig. 4f are the horizontal velocities for two vertical profiles (i.e., from 80 km in the lithosphere to 240 km in the asthenosphere). The profiles are located 100 km away from the plume stem. We rephrase these sentences.*

Lines 353-354: "However, without plume further supplies, the overpressure difference from the plume head to the spreading center decreases slowly with time (Fig. 355 8d)." What does it mean?

*Reply: Thanks for your comments. We reworded this paragraph in section 3.4.3 (lines 411-417)*

Lines 186-187: "The overriding plate moves slower than the ponding plume, and hence actually slows down the spreading plume branches." It is not clear what the message of this sentence is. According to model setup, since plume is located on the left side of MOR, the overriding plate motion speeds up the plume flow towards left (since the

plume flow and plate motion have the same direction) and slows down the flow in the right plume branch.

*Reply: We disagree with the reviewer. The left plume branch spreads in the same direction but faster than the overriding plate. Hence it is slowed down by plate drag. The right branch is moving in the opposite direction, and hence is also slowed down by plate drag. We improved the description of these processes (lines 235-239).*

Lines 187-188: "Without suction effect from the spreading center, the left plume branch flows out much slower than the right branch." Similar to what I mentioned in my previous comment, I expect faster flow towards left.

*Reply:We improved the sentences (lines 235-239). Please also see our reply to the previous comment. We also refer the reviewer to our added discussion of driving forces for plume spreading in section 3.3.*

Fig. 7: It is a very complicated figure. What do the upper panels of Fig. 7a-c stand for? They show the results at different times. Are the results shown in the lower panels of Fig. 7a-c showing the results at similar times as those shown in the upper panels? The scale of Fig. 7a-c is very small and one can hardly distinguish all the curves shown in the Figure. Please make the figure bigger. I suggest to move the legend of Fig. 7a into the right side of Figure because Fig. 7b-c also shows the results of models with different plume-ridge distances. The colors of curves for different plume-ridge distances are very similar and hard to differentiate them from each other. I suggest to change the colors. What do "plume head stage- positive spreading out" and "plume tail stage- passive flow driven by plate" mean? How are buoyancy fluxes calculated?

*Reply: We appreciate your suggestions and replot the Figure 7 (Fig.6 in the revised manuscript).*

[Figure]

**Figure 6** Models varying initial plume head radii (model M53, M58, and M63, Table S1 in supplementary material) shown by buoyancy flux and viscosity. **(a-c)** Buoyancy flux in spreading plume branches over time. Green and red triangles are markers used for buoyancy flux calculation. **(d-f)** Viscosity snapshots of models with different plume head radii. Models with green circle represent trench-ward flow and ridge-ward flow in red.

Line 330: How does Fig. 8a indicate that fast-spreading ridge promotes plume dragging. In this figure, from three models with fast spreading rates two are representing ridge suction mode.

***Reply:*** *We appreciate this comment. We replot the Figure 8 (Fig.9 in the revised manuscript) to demonstrate the effects of spreading rates, choosing different typical models. In fast spreading ridge models, more plume material is dragged away, which make it difficult for plume to interact with the ridge. Therefore, we suggested that fast-*

*spreading ridge promotes plume dragging.*

[Figure]

**Figure 9.** Model results influenced by different half spreading rates. **(a)** Effect of spreading rate on ridge-ward flow verse trench-ward flow. Viscosity snapshots are shown (model M7-M9, M82-M84, Table S1 in supplementary material). Fast-spreading ridge promotes plume material dragged. Models with green circle represent trench-ward flow and ridge-ward flow in red. **(b)** Dynamic evolutions of ridge-ward and trench-ward plume flow, revealed by defined ridge spreading fraction (eq.14). **(c)** Shear force (*Fs*) between moving plate and plume material under different spreading rates. **(d)** Pressure gradient between plume head and ridge center in different half spreading rate models. The solid and dash lines are the plume gravitation and dynamic pressure gradient, respectively.

Lines 340-347: It is not clear how shear force and pressure difference were calculated. Please re-write this part. Was the shear force calculated for the grids in the upper part of plume head or the whole plume head? The box of 50*50 km^2 in Fig. 8a is shown only for the plume head (and not for ridge center).

*Reply: Thanks for your advice. The shear force $F_s$ is calculated by integrating the*

*shear stress $\sigma_{xz}$ of the uppermost plume head grids, not the whole plume head. Moreover, we replace the pressure difference with pressure gradient to clarify the mechanism of ridge suction. We calculate the plume gravitation and dynamic pressure gradient by tracing the plume markers which record the pressure, density, etc. The method how we compute the shear force and pressure gradient have revised and described in detail (lines 301-308; lines 401-407).*

Fig. 8: What is the distance of plume-ridge in models shown in Fig. 8b-d? What do the dashed color curves in Fig. 8b stand for? Please explain them in the caption. The scale of figures are small. What does "plume head spreading" in Figure 8d mean? What is the effect of plume size on shear force and overpressure difference?

***Reply:*** *We appreciate your suggestions. We replotted the Figure 8(now Fig. 9 in the manuscript) and revised the caption. First, the "plume head spreading" in both figure and main text means the plume head spreads lateral under the plate. We explain it in the revised manuscript. Second, we added figures of shear force and pressure gradient of different plume size models in the supplementary material. The results shows that bigger size plumes are subjected to bigger shear force (Fig. S7). Meanwhile, bigger plume size means a more buoyant strong plume, which creates a larger pressure gradient between the plume and the ridge (Fig. S8). We have discussed this in the section 3.4.1.*

[Figure]

**Figure S7.** Shear force (*Fs*) between plate and plume in different plume head size models. The shear force imposed on the plume increases with plume size. The negative shear force indicates stronger friction imposed on the ridge-ward flowing (right) plume branch than that on the trench-ward flowing (left) plume branch.

[Figure]

**Figure S8.** Pressure gradient between plume head and ridge center. The plume gravitation and dynamic pressure gradient of **(a)** different size plumes, **(b)** different plume-ridge distances are shown by solid and dash lines, respectively.

Lines 360-361: "while all models gradually switch from ridge suction in the plume head stage to dominant plate drag in the plume-tail stage" Is it valid for all models or only those representing plate drag regime?

*Reply: We appreciate this comment which triggers us a lot of thinking. The description may be incorrect here. Only those plate drag models shift from ridge suction in the plume head stage to dominant plate drag in the plume-tail stage. We removed this sentence to avoid semantic ambiguity.*

Figure 9: What do "MAR" and "EPR" stand for? Please explain them in the caption. How did the authors obtain the plume buoyancy flux (which indicate plume size) of hotspots shown in Fig. 9?

*Reply: The "MAR" and "EPR" indicate the Mid-Atlantic Ridge and the East Pacific Rise, respectively. The plume buoyancy flux data in Fig.9 come from Hoggard (2020) and are presented in different circle sizes. We modified the Figure 9 (Figure 10 in the revised manuscript) and revised its caption.*

---

## Author Comment (AC2)

Dear referee,

We are glad to receive the review report and would like to express our sincere thanks to you and the editors. Without the constructive comments from you, the quality of this manuscript cannot be significantly improved. All the comments and suggestions have been carefully considered to revise the manuscript. Detailed reply to all comments and the associate manuscript modifications are given below.

**Reply on RC2**

**1.** Please revise the language using a professional editor or native speaker. There are many areas that may cause confusion as they are currently written.

**Reply:** We appreciate your advice and have revised the manuscript carefully to improve the language, with the help of all the co-authors.

2. Definitions: one of the major problems with this work at the moment is the lack of a definition of "ridge suction". Plate drag is reasonably well explained as the frictional force imposed upon the sub-lithospheric plume material, but ridge suction seems to be simply anything that causes plume material to travel toward the ridge. Currently, I have to infer this definition since no clear description is given and the quantitative assessment of ridge suction is a fractional number looking at the volume of plume material flowing toward and away from the ridge. If ridge suction is only assess in this way, then this is inconsistent with the literature and should be retermed as ridgeward flow or something similar. Better, I believe the authors need to reassess their model results with a more consistent definition of ridge suction that can be quantitatively assessed.

**Reply:**

Thanks for your suggestions. The concept of "ridge suction" refers to Niu (2004), who suggest that the spreading ridge sucks the material from depths due to the pressure gradient between the ridge center and deep hot material. The buoyant mantle plume from deep is overpressured, while the ridge center is in the state of underpressure. Therefore, the ridge suction we termed in the manuscript indicates the dynamic pressure gradient between the plume and ridge, which drives the plume material flowing to the ridge. In any cases, we reworded most of the occurrence of "ridge suction" in the manuscript, because it is only one of the driving mechanisms for ridge-ward plume flow.

On the other hand, the active gravitational spreading of plume also contributes to the ridge-ward plume flow. We discussed and compared the gravitational gradient and ridge-ward dynamic pressure gradient of these two mechanisms in the main text (see section 3.3)

**3.** In the abstract, the authors claim that plate drag has not been studied very much. However, I think this is perhaps an overly strong statement. There are several studies that incorporate the affect of plate motion (and the consequent drag) on plume spreading including Ribe et al. (1995); Ribe (1996); Ribe and Delattre (1998); Ito et al. (1997), Hall et al., 2003; 2004, etc. Each of these works (and others) incorporates the affects of plate drag on plume spreading in their calculations. I think the authors should be clear about what aspect of their work contributes something these other authors do not.

**Reply:** We appreciate this comment. We revised and improved the description of the motivation for our study and its novelty in the abstract and introduction.

**4. Model Comments**

4.1 In my opinion, for the scope of this work, the model used is overly complicated and in some respects inaccurate for a mid-ocean ridge setting. For example, the authors are examining the flow of mantle material beneath a lithosphere and have included a 1.5 km thick sediment layer across the model, but near ridge (especially fast ridges) there is little to no sediment. In fact, even along the slow spreading MAR, 1.5 km of sediment does not occur along the ridge axis and, indeed not for a reasonable distance away. **Reply:** We appreciate this comment. In our models, we imposed an average 1.5 km sediment layer vertically following the typical oceanic lithospheric structures. Horizontally, we use a uniform thickness lithosphere. We agree to the reviewer that the sediment near the ridge is negligible. Indeed, the processes of erosion/sedimentation are also considered in our simulations. We plot the thickness of sediment in the reference models (Figs. S1, S4). We are confident that the additional sediment layer thickness does not affect the conclusions of our study.

**Figure S1.** Reference model (M12, see Table S1, same as Fig. 3) evolution of ridgeward plume flow shown by (a) crust and sediment thickness, (b) normal stress. The mantle plume weakens the overlying oceanic plate and changes the stress state of the overlying oceanic plate. Molten plume material beneath the lithosphere is extracted to the crust.

---

## Author Comment (AC3)

Dear referee,

We are glad to receive the review report and would like to express our sincere thanks to you and the editors. Without the constructive comments from you, the quality of this manuscript cannot be significantly improved. All the comments and suggestions have been carefully considered to revise the manuscript. Detailed reply to all comments and the associate manuscript modifications are given below.

**Reply on RC1**

1. The language of the manuscript needs to be fully checked and revised by a professional editing service or a native speaker

**Reply:** We appreciate your advice and have revised the manuscript carefully to improve the language, with the help of all the co-authors.

2. In this study, the relation between spreading rate and age of oceanic lithosphere is ignored. Usually, higher spreading rates create younger lithospheres at a constant distance. In this study, the authors assumed that the lithospheric age is constant near the side boundaries (50 Myr). As a result, by imposing some higher velocities near the side boundaries to simulate higher spreading rates, the lithosphere becomes under extension and since the ridge is the weakest point in the system the width of ridge changes (it is clearly seen in e.g., Figs 3e,4e and 5a-b); higher rates lead to wider ridges. Could the authors explain to what extend is this assumption realistic? I think the formation of cracks in the lithosphere is the consequence of this assumption

**Reply:** We appreciate this comment which triggers us a lot of thinking. Indeed, the initial temperature distribution of the oceanic plate consists of the half-space cooling model and thermal equilibrium part. The half-space cooling model is used to describe the oceanic plate younger than 50Myr, and the thermal equilibrium structure is used to describe older oceanic parts. The thermal equilibrium thickness of the older lithosphere is constant (i.e., ~100 km; corresponding to a thermal age of 50 Ma). We further checked our model results, especially the stress state in the whole lithosphere. The result

shows that the lithosphere seems under extension owing to the high internal velocities (Figs. S1, S4). Actually, the "tension cracks" used in the main text may be inaccurate. There is no weakening or plastic deformation in the lithosphere, and there are no normal faults near the surface in the models. We describe the stress distribution in the lithosphere to highlight the stress localization, which occurs in the lithosphere where plume flows to the ridge. As a result, we revised and simplified the description of the "cracks" in the main text (lines 212-213 and lines 249-250).

**Figure S1.** Reference model (M12, see Table S1, same as Fig. 3) evolution of ridgeward plume flow shown by (a) crust and sediment thickness, (b) normal stress. The mantle plume weakens the overlying oceanic plate and changes the stress state of the overlying oceanic plate. Molten plume material beneath the lithosphere is extracted to the crust.

---

## Author Comment (AC4)

Dear referee,

We are glad to receive the review report and would like to express our sincere thanks to you and the editors. Without the constructive comments from you, the quality of this manuscript cannot be significantly improved. All the comments and suggestions have been carefully considered to revise the manuscript. Detailed reply to all comments and the associate manuscript modifications are given below.

**Reply on RC2**

**1.** Please revise the language using a professional editor or native speaker. There are many areas that may cause confusion as they are currently written.

*Reply: We appreciate your advice and have revised the manuscript carefully to improve the language, with the help of all the co-authors.*

**2.** Definitions: one of the major problems with this work at the moment is the lack of a definition of "ridge suction". Plate drag is reasonably well explained as the frictional force imposed upon the sub-lithospheric plume material, but ridge suction seems to be simply anything that causes plume material to travel toward the ridge. Currently, I have to infer this definition since no clear description is given and the quantitative assessment of ridge suction is a fractional number looking at the volume of plume material flowing toward and away from the ridge. If ridge suction is only assess in this way, then this is inconsistent with the literature and should be retermed as ridgeward flow or something similar. Better, I believe the authors need to reassess their model results with a more consistent definition of ridge suction that can be quantitatively assessed.

**Reply:**

*Thanks for your suggestions. The concept of "ridge suction" refers to Niu (2004), who suggest that the spreading ridge sucks the material from depths due to the pressure gradient between the ridge center and deep hot material. The buoyant*

*mantle plume from deep is overpressured, while the ridge center is in the state of underpressure. Therefore, the ridge suction we termed in the manuscript indicates the dynamic pressure gradient between the plume and ridge, which drives the plume material flowing to the ridge. In any cases, we reworded most of the occurrence of "ridge suction" in the manuscript, because it is only one of the driving mechanisms for ridge-ward plume flow.*

*On the other hand, the active gravitational spreading of plume also contributes to the ridge-ward plume flow. We discussed and compared the gravitational gradient and ridge-ward dynamic pressure gradient of these two mechanisms in the main text (see section 3.3)*

**3.** In the abstract, the authors claim that plate drag has not been studied very much. However, I think this is perhaps an overly strong statement. There are several studies that incorporate the affect of plate motion (and the consequent drag) on plume spreading including Ribe et al. (1995); Ribe (1996); Ribe and Delattre (1998); Ito et al. (1997), Hall et al., 2003; 2004, etc. Each of these works (and others) incorporates the affects of plate drag on plume spreading in their calculations. I think the authors should be clear about what aspect of their work contributes something these other authors do not.

*Reply: We appreciate this comment. We revised and improved the description of the motivation for our study and its novelty in the abstract and introduction.*

4. Model Comments

4.1 In my opinion, for the scope of this work, the model used is overly complicated and in some respects inaccurate for a mid-ocean ridge setting. For example, the authors are examining the flow of mantle material beneath a lithosphere and have included a 1.5 km thick sediment layer across the model, but near ridge (especially fast ridges) there is little to no sediment. In fact, even along the slow spreading MAR, 1.5 km of sediment does not occur along the ridge axis and, indeed not for a reasonable distance away.

*Reply: We appreciate this comment. In our models, we imposed an average 1.5 km sediment layer vertically following the typical oceanic lithospheric structures. Horizontally, we use a uniform thickness lithosphere. We agree to the reviewer that the sediment near the ridge is negligible. Indeed, the processes of erosion/sedimentation are also considered in our simulations. We plot the thickness of sediment in the reference models (Figs. S1, S4). We are confident that the additional sediment layer thickness does not affect the conclusions of our study.*

[Figure]

**Figure S1.** Reference model (M12, see Table S1, same as Fig. 3) evolution of ridge-ward plume flow shown by (a) crust and sediment thickness, (b) normal stress. The mantle plume weakens the overlying oceanic plate and changes the stress state of the overlying oceanic plate. Molten plume material beneath the lithosphere is extracted to the crust.

[Figure]

**Figure S4.** Reference model (M77, see Table S1, same as Fig. 4) evolution of trench-ward plume flow shown by (a) crust and sediment thickness, (b) normal stress. The mantle plume weakens the overlying oceanic plate and changes the stress state of the overlying oceanic plate. Molten plume material beneath the lithosphere is extracted to the crust.

4.2 Next, why is melting and heat flux useful for this study? Given the stated goal of the study to assess plume flow, I do not see (and it was not stated) why melting was useful or necessary. It is also unclear how the movement of melt throughout the system does or does not violate conservation of mass since you are working with an incompressible material and claiming to add material beneath the crust after removing it from another location. Please justify the use of melting and melt movement/accumulation. Also, clearly state any affect this melt has on your model (viscosity? temperature structure? density?, etc.)

*Reply: We appreciate your comments. In the model, melting of plume and asthenosphere are taken into account. The molten plume material is transformed to mafic magma and added to the crust, forming a thickened crust. We discussed the thickened crust between the plume and ridge to indicate the formation of oceanic plateau near the ridge during plume-ridge interaction. Besides, the melt extraction in the model is mass-conservative, as has been applied in previous modeling works* (Gerya et al., 2015; Gülcher et al., 2020). *We have modified the description of the melt extraction and its mechanism in the method section (lines 134-158). We also agree that the heat flux may not be that important for our study, so we remove the discussion of heat flux in the revised manuscript.*

4.3 Why do you need a plastic rheology? Given the scale of the problem you are working on, is the added focusing of the ridge axis to a smaller number of grid cells necessary? I don't believe that the current models can answer this question given the problem I mention next.

*Reply: Thanks for your comments. we employ viscoplastic rheology in the models. However, it is true that the stress and strain of ridge area usually are not strong enough to exceed the yielding criterion. The lithosphere does not exhibit plastic deformation in the model. Since plastic deformation does not occur in a widespread manner in our models, it indeed does not seem to be critical to be included in the model. On the other hand, including it for completeness (as we do here) does not change the main results of our study.*

*Secondly, we added smaller grid cells (grid size decreases linearly from 20km at the edges to 2 km at the ridge axis). Indeed, we set denser grids in the middle model domain in order to better simulate the interaction between plume and ridge. Of course, considering the scale of our research problem, the size of model grid does not have a great influence on the results. According to initial test cases, using smaller grids helps to make the computation more stable and does not fail to converge due to numerical perturbations.*

4.4. Another issue is the lack of adjustment of the lithosphere for plate spreading rate. In other words, the ridge and lithosphere in the "fast spreading, ridge drag dominated" cases do not appear to be in equilibrium before the plume is introduced. Looking that the compositional slices and temperature contours of the model in Figure 4, it appears that the sub-axial lithosphere is flattening out and a new, flatter lithosphere is forming without the initial half-space cooling structure (or with one that is in equilibrium with the faster spreading rate). This will alter the mantle flow field, the upslope topography of the ridge, and, potentially the location of the spreading ridge.

*Reply: We appreciate this comment and further checked our models. According to the model viscosity and temperature structure (Figs. S5, S6), we think that the flattening of the 1300 ℃ isotherm beneath the MOR is caused by the latent heat consumption during asthenospheric melting beneath the ridge. The effect of latent heating is not included in the initial temperature structure. Thus, while it is true that the 1300 ℃ isotherm flattens, the lithosphere does not flatten. It behaves as expected (see plots of the viscosity filed; Figs.S3, S6). Accordingly, we are confident that our predicted temperature and viscosity structure beneath the MOR is realistic.*

[Figure]

**Figure S5.** Temperature evolution of reference trench-ward plume flow model (M77, see Table S1, same as Fig. 4).

[Figure]

**Figure S6.** Viscosity evolution of reference trench-ward plume flow model (M77, see Table S1, same as Fig. 4).

4.5. Thermal structure

4.5.1 It is not clear to me how you arrive at a bottom boundary condition of 2513K when the base of the lithosphere has a Tmax = 1573. Since the base of the lithosphere is at ~100 km depth (Figure 1) and there is an imposed 0.5 K/km adiabatic temperature gradient, the max temperature at 660 km depth should be 1573 K + 560km*0.5K/km = 1853K. This is a big discrepancy that might imply a much hotter mantle than is realistic, which would likely have significant impacts on the results of this study.

*Reply: The model sizes in our study are set as 6600(width) and 1200(depth). The temperature of 2513K refers to the temperature at the bottom of model, that is, the temperature at 1200 km. Based on adiabatic temperature gradient of 0.5 K km⁻¹, the temperature at 660 km in all models is set initial to 1573 K + (660-120) km*0.5K/km*

*=1843K.*

4.5.2 How is the plume tail maintained? This is not clear or perhaps I missed it

**Reply:** *Thanks for your comments. The mantle plume imposed in our model is activate by given excess temperature at the beginning. Indeed, this temperature anomaly is only an initial condition, but not a long-term boundary condition. We do not impose a hot patch at the base of model, which means the mantle plume can uprise or erupt only once and does not create a long plume tail self-consistently. Considering that hotspots erupt periodically on earth, at intervals of millions of years. Therefore, in our study, we simplified the formation of mantle plume only considered one pulse of plume uprising. Such plumes are also widely used in similar studies* (Baes et al., 2016; Gerya et al., 2015; Gülcher et al., 2020).

5. Results/Interpretation

5.1 The images in Figure 4 demonstrate a factor that may explain the affect of plume head size on ridgeward flow – the erosion of the lithosphere by the plume head. As pointed out by Kincaid et al., 1995 in their laboratory experiments, the formation of lithospheric levees can act to block plume flow. This appears to be happening here. Small plume heads eat into the lithosphere a bit, effectively create ridges (or levees) that are the same thickness as the plume material and halt its motion. Then, as the plate moves the plume material has no choice but to flow with the plate. In contrast, large plume heads push the lithosphere out of the way all the way to the ridge. Despite the significant ridgeward flow, I would argue that this has nothing to do with ridge suction, but the lithosphere rheology and plume buoyancy forces.

**Reply**: *We appreciate this comment and give the following reply. As mentioned in our reply to comment 2, we suggest that the dynamic pressure gradient from the high-pressure plume to the low-pressure ridge actually contributes to the ridge-ward plume flow. The pressure gradient from the plume head to spreading center is notable (Fig. 5). Such pressure gradient varies with plume buoyancy force and different off-axis*

*distances. An increase in the plume head size, indeed, enhances the dynamic pressure and promotes the plume flowing ridge-ward (Fig.S8a).*

*Primarily, the plume flows to the ridge owing to the pressure gradient at first, which decrease gradually when plume get closer to the ridge. Then, the gravitational spreading of plume starts to drive ridge-ward flow. As the reviewer points out, the gravitational gradient is related to topography at the base of the lithosphere. See section 3.3 for a detailed discussion.*

[Figure]

**Figure 5.** Comparsion between models with ridge-ward vs. plate-drag plume flow. **(a)** Ridge-ward flow with downwelling beneath the MOR (results from case M12 as in Figure 3). White dashed lines are streamlines; black arrows visualize the flow field. Schematic of flow in the sub-panel on the right-hand side. **(b)** Plate-drag flow with upwelling mantle corner flow beneath the MOR (results from case M77 as in Figure 4). **(c)** The dynamic pressure and gravitational gradient of plume marker (i.e. green circle in (a)) over time. The yellow box in (b) marks the location for the computation of average dynamic pressure at the ridge, needed for the calculation of the dynamic

pressure gradient (see text). **(d)** The dynamic pressure and gravitational gradient of plume marker (i.e. green circle in (b)) over time.

5.2. The claim of "tension cracks" seems to be based on the stresses in the model. These stresses reach maximums of + or – 3x10^-7 Pa (Figure 5), much too small to actually fracture of rock - especially near the surface, which typically has yield strengths many orders of magnitude larger. Is this a typo? If this should be + or –3x10^7 Pa (i.e., 30 MPa) that seems very large and so I am left to question how tension cracks are justified here. However, I would note that I don't think these are essential for the results of this paper and fall into the over complication of the model for the state purpose of the modeling.

*Reply: Thanks for your comments. Yes, the magnitude of normal stress here is a typo. The correct magnitude of these stresses should be $\pm 3 \times 10^7$ Pa in the models. Actually, the "tension cracks" used in the main text may be inaccurate. There are no brittle fractures in the lithosphere and normal faults near the surface in the models. While there are horizontal extensional stresses, there is in fact no yielding. As a result, we removed the description of the "cracks" in the main text.*

5.3. The role of ridge suction vs plate drag. I think the authors have glossed over some of the factors likely to contribute to the plume flow including the slope of the lithosphere and its role in guiding plume material up the slope, the buoyancy flux of the plume stem since this is not described by the plume radius definition here (which seems to describe the size of the plume head).

*Reply: We appreciate this comment which triggers us a lot of thinking. We think the effect of the lithospheric slope is relative to the half spreading rate. The slope of the lithosphere varies with the spreading rate of the mid-ocean ridge. The base of the lithosphere would be flatter at the fast-spreading ridge. Consequently, a slow*

*spreading ridge imposes smaller shear force on plume head and forms a steeper lithosphere base which benefits to the ridge-ward plume flow. Besides, we now evaluate the plume gravitational gradient, considering the local lithospheric slope, and ridge-ward dynamic pressure gradient (Fig. 5). For a detailed analysis, see section 3.3.*

5.4. Much of the interpretation of these results hinge around spreading of the plume head, not the plume after it has established itself beneath the lithosphere. Many plume have been active for 10 Myr or more and the plume head will have greatly diminished or completely spread away by that time. Yet, these plume tails can still interact with ridges since ridges migrate and often approach plumes. How does the long term interaction look - after the plume head has disappeared?

*Reply: According to our results, the plume tail (which follows the plume head at model times 1~8 Myr) bends either toward or bend away from the ridge in ridge-ward flow and plate-drag flow dominated models, respectively. Taking the long-term evolution of the model, the degree of plume tail tilts towards or away from ridge increases with time, and ends up maintaining a relative steady state. Please also see our previous reply to 4.5.2.*

5.5. Related to 4., I don't think the authors should be claiming to assess plume radius, as this commonly is used to refer to the radius of the plume stem. Instead, I think the manuscript would be much clearer if the authors would state that they were varying the plume head radius.

*Reply: We appreciate this comment and made revisions. In this study, we varied the plume buoyancy flux by changing its radius. The plume radius in our model only refers to the initial size of mantle plume. The width of plume stem decreases dramatically when the plume rising to the plate and plume head spreading out, which is different from the columnar plume stem radius detected by the geophysical tomography. Thus, as you suggested, we use the "initial plume head radius" to replace "plume radius" in the main text.*

**Other minor specific comments:**

•Line 47-48 – I'm not clear as to what this statement has to do with the EPR sucking in plumes so that they do not appear near the ridge.

*Reply: Thanks for your comments. We removed this sentence and reword this paragraph in introduction section.*

•Line 52 – the use of the work "push" is inappropriate here and should be changed to "drag" or similar

*Reply: Thanks for your advice. We replace the "push" with "convey" in the text (line 54-55).*

•Line 58 - The authors should reconsider how they phrase things – for example, "slow spreading rate, short distance (small plume-ridge distances??), and large plume radii promote ridge suction,…" is an inaccurate statement – really, I think what the authors are trying to say is that these factors favor plumes being pulled toward ridges by ridge suction

*Reply: Thanks for your advice. We rephrased all these inaccurate phrases in the main text.*

•Line 59 – maybe try a more careful wording – it is the fast plate motions associated with fast-spreading ridges that exert strong drag forces on plumes

*Reply: Thanks for your advice. We revised this paragraph to clarify our goal of this study in the main text.*

Figure 2 – this does not look like a half-space cooling model. Is this a plate cooling model or some modified half-space model? The half-space cooling model does not flatten like this.

*Reply: We appreciate this comment and made revisions in figure 2 and the main text. Indeed, the initial temperature distribution of the oceanic plate is prescribed by the half-space cooling model and thermal equilibrium structure. The half-space cooling*

*model is used to describe the oceanic plate younger than 50Myr, and the thermal equilibrium structure is used to describe older oceanic parts. We set up the models in this way because we consider that the theoretical half-space cooling model has a good match with some geophysical observations when the plate is young, but the fit becomes poor when the age is greater than 60/70 Ma (Turcotte and Schubert, 2014; Stein and Stein, 1994). Therefore, we set a half-space cooling model with a maximum age at 50Ma, and the thermal equilibrium thickness of the older lithosphere is constant (i.e., ~100 km; corresponding to a thermal age of 50 Ma).*

[Figure]

**Figure 2.** Model setup. **(a)** Initial composition and boundary conditions. The oceanic plate consists of half-space cooling part and the thermal equilibrium part. A 50-Myrs-old mid-ocean ridge sets in the middle of the model based on half-space cooling temperature structure. A thermal and chemical anormal mantle plume locates at 660 km. Different colors indicate the initial rock types and corresponding newly formed molten rock types. Yellow arrows are the half-spreading rates imposed internal in the lithosphere (i.e., from 20 km to 120 km in depth) to simulate ridge spreading. **(b)**

Initial tested ridge and plume configurations. (**c**) Initial tested plume-ridge distances.

---

## Author Response (AR2)

Dear Editor,

We are glad to receive the review report and would like to express our sincere thanks to you and the reviewers. Without the constructive comments from the reviewers, the quality of this manuscript cannot be significantly improved. All the comments and suggestions have been carefully considered to revise the manuscript. Detailed reply to all comments and the associate manuscript modifications are given below.

**Comments from the reviewer #1**

1. First, as I said in my initial review, I'm still confused as to why the model being used has so many seemingly extra processes unrelated to the problem at hand. For example, why was sediment and the processes of erosion and sedimentation included? Why was plasticity included if faulting and fracturing are not important to the system evolution? Why was melt transport and crustal accumulation included? My general question regarding the all of the above is "what is the goal of adding these processes and how do they affect (or not) the results?" Most importantly, in the current draft the authors do not demonstrate how the above processes affect their results. I suppose that it is possible that all of these are essential to the problem the authors are examining. Yet, even if they do not affect the results, I think the authors should demonstrate this to the reader – otherwise the paper ends up being difficult to interpret with certainty.

Reply: The reviewer questioned our model with "extra processes". We appreciate this comment, but we disagree with the reviewer. We disagree that a simplified model is better than a sophisticated model. Actually, the "extra processes" (e.g., sedimentation/erosion, plasticity, and melt processes) used in our model are the basic elements in the sophisticated numerical codes in the geodynamic community. Our mode is deal with large scale geodynamic processes (i.e., plume migration and mid-ocean ridge spreading), and some of the processes as the reviewer mentioned may not be the first-order controls, but they are important and one should not exclude. For instance, we keep sediments above the crust, because it represents the real geological background since the ocean crust is indeed covered by sediments.

Although we do not emphasize faulting and fracturing which are related to plasticity, we do see the effect of plasticity in the oceanic plate above the arising plume (through strain rate field). Furthermore, the processes of melt transport and crustal accumulation are important features of plume-lithosphere interaction in our models. Specifically, oceanic plateau or thickened crust is formed near the ridge when the plume flows to the ridge, whereas molten plume is hardly extracted when plate drags plume away.

2. With the melting process, equations 12 and 13 show that conservation of mass (volume) for an incompressible material is only being obeyed globally and not locally. Locally, on a given grid element or node, when mass is removed and moved elsewhere this will violate div(v) = 0 at that point. I think it would be useful for the author's to acknowledge this in their model set up and to discuss how using a compressible formulation might affect the solution (or not) and what evidence supports their assertion.

**Reply:** The melt extraction in the model is definitely mass-conservative, as has been applied in previous modeling works (Gerya et al., 2015; Gülcher et al., 2020).

3. The plume head appears to be nearly always pinned at the minimum viscosity of the model – could the authors comment on this and explain why they did not try larger plume viscosities (or lower plume T)? It seems that the viscosity contrasts are going to be very important for the behavior of the plume material.

**Reply:** Thanks for the advice. In fact, we had systematically tested the influence of plume temperature on plume-ridge interaction, and found that plume temperature affects model evolution in the following ways. Firstly, plume temperature influences the rate of plume rise. Secondly, higher plume temperature can increase the molten plume material, leading to more intense interaction between the mantle plume and the lithosphere, resulting in a lot of magmatic activity. But in general, increasing or decreasing the plume temperature does not directly change the model type in the parameter plot (Fig. 8).

**Figure 1.** Model evolution of ridge-ward plume flow with different plume temperatures shown by (a) crust and sediment thickness, (b) composition.

4. Figure 3b-a. Can the author's explain why the models exhibit a long lived plume tail. The author's describe the impetus for their plume as a semi-circular heat patch at 660 km depth. Yet, looking at figure 3 there is compositionally "plume" material for >8 Myr. I'm a bit surprised about this. Shouldn't the material have buoyantly risen above its initial depth by this point?

**Reply:** In our model setup, the plume initially exists as a semicircle heat patch, driven upward by the excess temperature. When the model evolves by 8 Myr, the whole mantle plume has risen below the plate, and a very small portion of plume material maintain in deep. This may be due to the deep mantle circulation that causes a little material remained in the deep. But the total amount of plume material is conserved throughout the evolution. There is no continuous supply of material from the deep mantle to the plume. In fact, as the model evolves over time, residual plume tail material will rise beneath the plate eventually.

5. A discussion of likely plume head size given our knowledge of mantle parameter space would be useful here. What is a likely radius of a plume head in the mantle? What would be the expected range?

**Reply**: We appreciate this comment. The radius of the plume in the mantle can be roughly measured from seismic tomography. Based on finite-frequency tomography, Montelli(2004) present evidence for the existence of deep-mantle thermal convection plumes and reveals the plume has a radius of 100 km to 400 km. Besides, in previous simulation studies on mantle plumes, the reference range of plume radius is also within the range of  $0 \sim 300$  km. As a result, We believe that the plume radius parameter used in the model is reasonable.

**Minor comments:**

Line 296: "massive melting and crust production" is very vague and not quantifiable. Please either replace with numbers from the model or simply state that melting and crustal production occur within the model.

Reply: We have made revision in the main text (line 219).

Line 582 – "gird" should be "grid"

Reply: We have made revision in the main text (line 414).

**Comments from the reviewer #2**

**1.** lines 479-481: "Therefore, the effects of plume size are not a good candidate to explain the notable difference between the Atlantic and Pacific in terms of plume-ridge interaction mode." There is a discrepancy between Fig. 1b and this sentence. Figure 1b shows that plume buoyancy is an important factor in ridge-ward or plate-drag motion of plume. Based on Fig. 1b, plumes with low buoyancy flux (small plumes) favours plume-ridge interaction. Is the data in Fig. 1b based on observations? Why does it differ from Fig. 11b which indicates negligible effect of plume buoyancy?

**Reply:**

Thanks for your suggestions. Indeed, the plume buoyancy flux data used in the fig.1b and fig.11b are the same. The buoyancy fluxes of mantle plumes refer to the Hoggard(2022), which is calculated from the topographic swell volume, and considers both the decay of buoyancy through time and the differential motion between asthenospheric buoyancy and the overlying plate. Based on Fig.1b, it is true that plumes with low buoyancy flux tends to interact with nearby ridge. However, it is also worth noting that these small plumes tend to be very close to ridge. Indeed, figure 11 shows that when the mantle plume is close enough to the ridge, even the mantle plume with small buoyancy can flow to the ridge. This also shows that apart from the buoyancy flux of mantle plume, distance also plays a very important role in the effect of plume-ridge interaction, which is consistent with our simulation results.

2. Lines 357-363: I believe cooling of the plume in the time scale of authors model (~7 Myrs) is not a good explanation for plate drag mode of plumes with large plume-ridge distance. Plume sustains hot in such a time scale as it can be seen in e.g. Figure 7. I think the main reason for plate drag in the case of distant plumes is the mantle plume flux which is not large enough to allow the plume materials to reach the ridge (due to the large plume-ridge distance). This is the same for small sized plumes. *Reply: Thanks for your comments. We agree with you. We believe that the first-order factor for plate drag models is the mantle plume flux or plume size, which we have*

discussed in section 3.4.1. Indeed, when plume uprises with large flux, plume material can still flow into the ridge even at a large plume-ridge distance (Fig.6). Indeed, we discuss the effect of thermal cooling of plume in section 3.4.2, aiming to reveal that the plume-ridge distance can also affect the plume-ridge interaction. Larger distance forms lower pressure gradient between the plume and ridge, and it takes longer time for plume flowing to the ridge, leading to the cooling of plume and plume viscosity increased. In fact, chances are that the plume may sustain hot in the first Myrs after ponding the lithosphere.

**Other minor specific comments:**

• Line 42: what is "on-axis and off-axis plume"

**Reply:** Thanks for your comments. The "on-axis and off-axis plume" mean ridge-centered and off-ridge plume. We reword this sentence in introduction section.

• Lines 60-61: "The distribution of hotspots with classified as plume-ridge interaction (ridge-ward spreading) vs. no interaction (plate-drag spreading) also still remains enigmatic. "This sentence is not clear.

**Reply:** Thanks for your advice. We reword this sentence in introduction section.

• Lines 64-65 : What is "horizontally propagating viscous finger"? I suggest to briefly explain it in paratheses here.

**Reply:** Thanks for your advice. We reword this sentence in the manuscript.

• Line 296: Replace "The" by "the"

Reply: We have made revision in the main text.

• Line 19: "Slow spreading rates" should be added here.

Reply: We have made revision in the main text.

• Line 20: "Large plume-ridge distance" should be added here.

Reply: Thanks for you comments. We have made revision in the main text.

Line 398: "The transition from ridge-ward (positive γ) to plate-drag (negative γ) flow" In Figure 9, the orange curve (with spreading rate of 30 cm/yr) has positive γ but its deformation mode (based on Table S1) is plate-drag.

**Reply:** The orange curve in fig. 9 (M58: half spreading rate of 30 mm yr-1, an initial plume head radius of 150 km, and an off-axis distance of 1000 km) should be classified as plate drag flow. The spreading fraction  $\gamma$  is positive at first, but turns to negative rapidly due to drag of moving plate. Actually, there is no plume material flowing into the ridge.

• In Fig. 5 thickness of lithosphere is more than 200 km but in Figs. 3-4 lithosphere is thinner!

**Relpy:** Thanks for your comments. The Y-axis tick in Fig.5 were labeled incorrectly. We have modified the tick in Fig.5.

• Line 527: It should be fast spreading rates.

**Relpy:** Thanks for your comments. We have made revision in the main text.

---

## Author Response (AR3)

Dear Editor,

We are glad to receive the review report and would like to express our sincere thanks to you and the reviewers. Without the constructive comments from the reviewers, the quality of this manuscript cannot be significantly improved. All the comments and suggestions have been carefully considered to revise the manuscript. Detailed reply to all comments and the associate manuscript modifications are given below.

***Comments from the Anonymous Referee #1***

Line 21: I suggest to remove "at least for".

***Reply:*** *Thank you for your comments. We revised this sentence (line 21).*

Lines 189-190: I still believe that thickness of 100km for oceanic lithosphere of 50 Myr is unrealistically large.

**Reply:** *We appreciate this comment. In our models, we imposed a constant initial 100 km lithosphere for plate older than 50Myr. We agree to the reviewer that the lithosphere thickness is significant large, based on the standard half-space cooling model. However, if the entire model uses half cooling space, the half-space cooling width (spreading rate times plate age) will be very large, far beyond the size of the model. So, given the initial setup of the model, we set the initial temperature distribution of the oceanic plate, consisting of the half-space cooling model and thermal equilibrium part. On the other hand, the plume-ridge interaction in our models mainly occurs between the plume and ridge, where the lithosphere temperature structure follows the standard half-space cooling model. We believe that the thickened lithosphere does not affect the conclusions of our study.*

Line 210-211: You have to keep the sequential order of figures in the text (first Fig. 3a then Fig. 3b). The same is for line 362.

***Reply:*** *We appreciate your advice and have revised the manuscript carefully.*

Line 219: "As the plume eventually flows upward along the increasingly sloping base of the plate near the MOR" This part is not clear, please re-write it.

*Reply: We improved the sentences (lines 218-220).*

Line 226: I suggest to remove this sentence.

*Reply: We appreciate this comment. We removed this sentence in the text.*

Lines 240-242: The branches of the spreading plume head move significantly faster than the overriding plate. Therefore, plate drag actually slows down the spreading of the plume branches in this model case" I do not quite agree with this part. Since velocity of plume materials are much higher ( > 10 cm for right and ~ 5 cm for the left branch) than plate drag (8 mm/yr), therefore, one can conclude that the effect of plate drag is negligible in this case. One can also conclude that higher velocity of spreading plume may slow down the plate drag. That is an interesting issue which can be investigated by looking at the velocity of the overriding plate before and after plume uprising (to see any changes).

*Reply: We agree with you and removed this sentence. Indeed, the velocity of plate in the models do not change over time by imposing constant internal boundary velocities in the lithosphere. The velocity of plume branches are much higher than plate, which may slow down the plate drag in turn. We appreciate this comment which triggers us a lot of thinking. The condition that triggers spreading plume to slow down the plate remains intriguing but unclear. More experiments are required to test to investigate the mechanism in the future.*

Lines 255-256: "divergent stresses are sustained in the overlying lithosphere (Fig. S4), but no weakening or yielding occurs (Fig. S6)." What do you mean by divergent stress? In Fig. S4, one sees the change in the stress in the lithosphere which indicates that the lithosphere is weakened (it is also indicated in the figure caption).

*Reply: Thanks for your comments.* The buoyant plume changes the stress state of the overlying oceanic plate when it interacts with the lithosphere. The divergent stresses here means that the stress state in the lithospheric region affected by mantle plume changes, compared with the normal lithospheric stress state. *We have made revision in the main text (lines 252-253).*

Line 258: "However, thick and cold lithosphere prevents magma from extracting (Fig. S4)." The lithosphere in Fig. 3 is even colder and thicker (as the spreading velocity is lower). The reason here is the low melt flux due to small plume.

*Reply: Thanks for your advice. We revised this sentence (line 255).*

Figure caption of Fig. 5 c and d: Please indicate that the(c) and (d) are representing the results of ridge-ward and plate-drag flow, respectively. Is the average dynamic pressure the same in both model (I do not think so)? Please show the location of yellow box in Fig. 5a as well and modify the caption accordingly.

*Reply: Thanks for your advice. We have made revision in the main text (line 285).*

Lines 326-327: The last sentence of this paragraph is not clear. Please re-write it. Besides, you did not explain 5d in the text. Please explain this Figure in the text as well.

*Reply: Thanks for your advice. We have improved this sentence in the main text (lines 325-326).*

Figure caption of Fig. 6: Please indicate in the caption that U and R stand for half of spreading rate and plume radius. Please do the same for Figures 7 and 8.

*Reply: We appreciate your advice. We have made revision in the main text.*

Line 355: Please modify this part as: "Green and red triangles are markers used for

buoyancy flux calculations of left and right plume branches, respectively." Please do the same for Figs. 7 and 8.

*Reply: Thanks for your advice. We have made revision in the main text.*

Fig S8: What are the dashed curves? Please explain them in the caption.

*Reply: Thanks for your advice. We revised the caption in the supplementary material.*

Lines 365-371: I suggest to remove this part. Because there is not strong evidence for that in your models (they do not show any reduction of viscosity with time). Moreover, all of your models (both ridge-ward and plate-drag) show decrease of plume temperature in time.

*Reply: We appreciate this comment and agree with the reviewer. We removed this part in the main text.*

Figure 8: In this figure only models of Figure 9a are shown. Please also show the models of Figure 9b-d.

*Reply: We appreciate your suggestion and replot the Figure 8.*

Figure caption of Fig. 9: This figure shows the effect of half spreading rate and plume-ridge distance on ridge-ward vs plate-drag motion. Please correct the caption.

*Reply: Thanks for your advice. We have revised the caption in the main text.*

Line 429: "Fast-spreading ridge promotes plume material dragged " This is true for plumes near the ridge. Please correct it.

*Reply: Thanks for your comment. We have made revision in the main text (line 425).*

Lines 446-447 : "We discuss the viability of this potential explanation by comparing with geological and geophysical observations (Fig. 10). " Please re-write this sentence.

*Reply: Thanks for your comment. We have reworded this sentence in the main text (lines 446-447).*

Lines 481-483: I cannot understand what is the message of this part. Please rephrase this part and write it in more clear way.

*Reply: Thanks for your comment. We have made revision in the main text (lines 479-482).*

Lines 503-506: "In this case, plume-ridge distance may play a critical role in the plume-ridge interaction, and could explain the striking difference between the Pacific and Atlantic in terms of the number of plume-ridge interacting vs. non-interacting systems" This part of text is a bit unclear. Looking at Fig. 11b, I see that 4 plumes in Atlantic and 6 plumes in Pacific are moving towards the ridge. However, the total number of plumes in Pacific is higher than that in Atlantic. Figure 11b shows that a lager fraction (percentage) of plumes in Atlantic ocean is interacting with ridge compared to that in Pacific ocean. In other words, the comparison of absolute numbers of plumes in Atlantic and Pacific is not a good criterion here. It is better to consider the ratio of plumes moving towards ridge and total number of plumes. Please modify the text to clarify this issue.

*Reply: We appreciate this suggestion and agree with you. It is better to consider the proportion of plumes moving towards ridge and total plumes. Actually, A small minority of Pacific plumes (6 of 16), and a large majority of Atlantic plumes (7 of 14) display interaction with the ridge (Fig. 11a). We have made revision in the main text (lines 449-450,501-504).*

Line 523: Correct: "mark"

*Reply: We have made revision in the main text (line 521).*

Lines 525 and 526: Correct "comes"

*Reply: We have made revision in the main text (lines 523-524).*

Since the term "Bouyanc flux" is very important, it is better to define or clarify it somewhere in the text.

*Reply: Thanks for your comments. We have made revision in the main text (lines 339-340).*

Lines 107-109: Markers are not involved in solving these PDEs you list here. All of these PDEs are solved in the Euelrian grids. To avoid misunderstandings, you might need to rewrite this part.

*Reply: Thanks for your advice. We reword this sentence in method section (lines 107-109).*

Lines 172-173: It would be better to indicate the vertical resolution as well.

*Reply: We appreciate your advice. We have made revision in the main text (line173).*

Figure 1(a) In the bathymetry map, a lot of places are much higher than 0 m. I guess your color map is not right. Please check it.

*Reply: Thank you very much for your comments. Indeed, the bathymetry map in 1(a) is residual bathmetry, which is generated by subtracting the oceanic lithosphere subsidence surface from observed depths. Positive is anomalously shallow, and negative is anomalously deep.*

In Figure 6, 7,9, For the legend of colormap, it is better to use "viscosity" instead of "nu"

*Reply: Thanks for you comments. We have replotted the Fig.6,7,9 in the main text .*

Figure 10, It is unclear what the blue arrow means

*Reply: Thanks for your advice. We revised the caption in the main text.*